# Multiplex imaging relates quantal glutamate release to presynaptic Ca$^{2+}$ homeostasis at multiple synapses in situ

Thomas P. Jensen[1], Kaiyu Zheng [1], Nicholas Cole[1], Jonathan S. Marvin [2], Loren L. Looger [2] & Dmitri A. Rusakov [1]

Information processing by brain circuits depends on Ca$^{2+}$-dependent, stochastic release of the excitatory neurotransmitter glutamate. Whilst optical glutamate sensors have enabled detection of synaptic discharges, understanding presynaptic machinery requires simultaneous readout of glutamate release and nanomolar presynaptic Ca$^{2+}$ in situ. Here, we find that the fluorescence lifetime of the red-shifted Ca$^{2+}$ indicator Cal-590 is Ca$^{2+}$-sensitive in the nanomolar range, and employ it in combination with green glutamate sensors to relate quantal neurotransmission to presynaptic Ca$^{2+}$ kinetics. Multiplexed imaging of individual and multiple synapses in identified axonal circuits reveals that glutamate release efficacy, but not its short-term plasticity, varies with time-dependent fluctuations in presynaptic resting Ca$^{2+}$ or spike-evoked Ca$^{2+}$ entry. Within individual presynaptic boutons, we find no nanoscopic co-localisation of evoked presynaptic Ca$^{2+}$ entry with the prevalent glutamate release site, suggesting loose coupling between the two. The approach enables a better understanding of release machinery at central synapses.

[1] UCL Queen Square Institute of Neurology, University College London, Queen Square, London WC1N 3BG, UK. [2] Janelia Research Campus, Howard Hughes Medical Institute, Ashburn 20147 VA, USA. Correspondence and requests for materials should be addressed to T.P.J. (email: t.jensen@ucl.ac.uk) or to D.A.R. (email: d.rusakov@ucl.ac.uk)

Stochastic, $Ca^{2+}$-dependent release of the excitatory neurotransmitter glutamate by synapses underpins information handling and storage by neural networks. However, in many central circuits glutamate release occurs with a low probability and a high degree of inter-synaptic heterogeneity involving multiple aspects of synaptic organisation[1,2]. Therefore, understanding presynaptic function requires reliable monitoring of presynaptic action potentials (APs), presynaptic $Ca^{2+}$ dynamics, and release of glutamate quanta in individual synapses. The optical quantal analysis method went some way toward this goal by providing quantification of release probability at individual synapses in brain slices[3,4]. In parallel, advances in the imaging techniques suited to monitor membrane retrieval at presynaptic terminals have enabled detection of synaptic vesicle exocytosis in cultured neurons[5]. Recently developed optical glutamate sensors[6,7] have drastically expanded the sensitivity and the dynamic range of glutamate discharge detection in organised brain tissue[8,9]. However, such methods on their own cannot relate neurotransmitter release to presynaptic $Ca^{2+}$ dynamics, which is the key to unveiling presynaptic release machinery. Direct examination of this relationship has hitherto been feasible only in classical studies of giant synapses that permit direct electrophysiological access and probing of presynaptic $Ca^{2+}$ dynamics[10–12].

Furthermore, conventional imaging methods based on fluorescence intensity readout face multiple challenges when applied in turbid media such as organised brain tissue. Detected emission intensity can be affected by focal drift, photobleaching, or any experimental concomitants impacting on light scattering. To overcome such difficulties in $Ca^{2+}$ imaging, we have recently employed fluorescence lifetime imaging microscopy (FLIM) of the $Ca^{2+}$ indicator OGB-1[9,13,14]. The OGB-1 FLIM readout provided nanomolar-range $Ca^{2+}$ sensitivity, with little influence of light scattering, dye concentration, focus drift, or photobleaching. However, OGB-1 emission is chromatically inseparable from that of the existing glutamate sensors, thus prohibiting simultaneous imaging of glutamate release[9].

To deal with this challenge, here we systematically explore FLIM properties of $Ca^{2+}$ indicators and discover that the lifetime of red-shifted Cal-590, an indicator successfully used in deep-brain imaging[15], is also sensitive to low nanomolar $Ca^{2+}$. In parallel, we develop a glutamate sensor variant SF-iGluSnFR. A184S, which has a suitable dynamic range but a slower off-rate than the existing SF-iGluSnFR.A184V, and thus permits monitoring of glutamate release simultaneously at multiple sites. In organotypic hippocampal slices, we register concurrent, chromatically separated signals reporting evoked glutamate release and presynaptic $Ca^{2+}$ kinetics, at individual or multiple axonal release sites (recorded simultaneously or sequentially) traced from the soma. This multiplex imaging approach establishes some basic relationships between presynaptic resting $Ca^{2+}$, AP-evoked $Ca^{2+}$ entry, and synaptic release efficacy. At a sub-microscopic level of individual axonal boutons, we employ this method to evaluate localisation of glutamate release sites and how they are related to presynaptic $Ca^{2+}$ entry, an issue fundamental to neurobiology[16–18]. The present approach helps to understand better the mechanism of synaptic release, its use-dependent changes, and the underpinning $Ca^{2+}$ machinery in identified brain circuits in situ.

## Results

### FLIM of red-shifted Cal-590 enables low $Ca^{2+}$ readout. In search of suitable red-shifted $Ca^{2+}$ indicators, we tested and excluded the classical rhodamine-based dyes: they appear highly lipophilic, which would restrict imaging in axonal boutons away from the soma. In contrast, the fluorescein-based, red shifted

Cal-590 was previously found to provide reliable $Ca^{2+}$ monitoring for deep brain imaging in vivo[15]. We therefore set about testing the $Ca^{2+}$ sensitivity of its fluorescence lifetime, particularly for two-photon excitation, alongside the green glutamate sensor SF- iGluSnFR[6,7]. We found a clear dependence between the Cal-590 lifetime and $[Ca^{2+}]$ using a series of $Ca^{2+}$-clamped solutions, as described previously[13,14] (Fig. 1a; Methods). This dependence appeared strongest around $\lambda_x^{2P} \sim 910$ nm (Supplementary Figure 1a) and largely insensitive to temperature, hence associated viscosity, between 20 °C and 37 °C (Supplementary Figure 1b). Several other $Ca^{2+}$ indicators, including red-shifted Asante Calcium Red and Calcium Ruby-nano, showed no usable $[Ca^{2+}]$ sensitivity in their fluorescence lifetime (Supplementary Figure 1c).

Thus we used the protocol established for OGB-1 previously[13,14] to calibrate Cal-590 lifetime for $[Ca^{2+}]$. The procedure employs the normalised total count (NTC) method in which photon counts are integrated, as the area under the lifetime decay curve, over the $Ca^{2+}$-sensitive interval (~3 ns post-pulse for Cal-590), and the result is related to the peak fluorescence value (Fig. 1a, Methods). This ratiometric method substantially lowers the minimum requirement for the photon counts compared to traditional multi-exponential fitting in FLIM; this in turn shortens minimal acquisition time and improves spatiotemporal resolution of $Ca^{2+}$ imaging[13,14]. The Cal-590 FLIM calibration outcome showed the greatest sensitivity in the 0–200 nM range (Fig. 1b).

To test this approach in organised brain tissue, we turned to organotypic hippocampal slices. Because Cal-590 could not be reliably used for axon tracing (due to its weak emission at low $[Ca^{2+}]$), we employed the bright fluorescence of the optical glutamate sensor SF-iGluSnFR; two sensor variants, A184V and A184S, provided similarly bright expression labelling. We achieved sparse cell expression by biolistically transfecting SF-iGluSnFR (Methods), patch-loaded SF-iGluSnFR-expressing CA3 pyramidal cells with Cal-590 (300 μM), and traced their axonal boutons for at least ~200 microns from the soma, towards area CA1 (Fig. 1c). The axonal bouton of interest was imaged using a spiral ('tornado') scanning mode (Fig. 1c, bottom), which we showed previously to enable rapid (1–2 ms) coverage of the visible bouton area during single sweeps[9]. Next, we asked whether the dynamic range of Cal-590 FLIM readout inside axons in situ was similar to that in calibration conditions (Fig. 1a, b). We therefore recorded both intensity and lifetime decay of presynaptic $Ca^{2+}$ signal, first in low $Ca^{2+}$ resting conditions (for ~400 ms), and next following a 500-ms 100-Hz spike train evoked at the soma (Fig. 1d), which should reliably saturate the indicator[19,20]. Comparing the respective Cal-590 fluorescence decay curves confirmed a wide dynamic range of $[Ca^{2+}]$ sensitivity in situ (Fig. 1e), consistent with the calibration data in vitro (Fig. 1a, b).

### Multiple-site glutamate release monitoring. We next asked whether the kinetic features of the SF-iGluSnFR sensors were suited for simultaneous imaging of multiple axonal boutons. This was important because the laser beam has to dwell for 1–2 ms at each bouton to achieve signal-to-noise ratio comparable to that obtained under tornado scanning of individual boutons[9]. Thus, reliable imaging of fluorescent signals concurrently at 4–5 locations requires the signal to last (preferably near its peak) for at least 10–15 ms. Fortuitously, this requirement is normally met by high-affinity $Ca^{2+}$ indicators, including Cal-590[15].

We noted that our recently developed glutamate sensor variant SF-iGluSnFR.A184S[7] had a much lower off-rate than the faster variant SF-iGluSnFR.A184V (Supplementary Figure 2a). To test

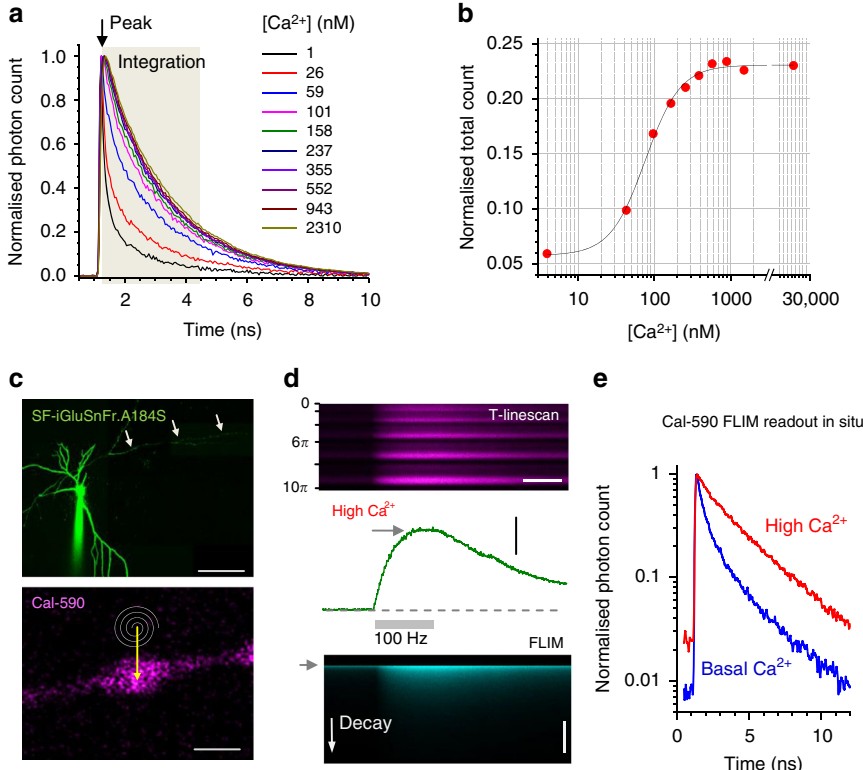

**Fig. 1** Fluorescence lifetime of Cal-590 provides readout of low intracellular $Ca^{2+}$. **a** Fluorescence lifetime decay curves of Cal-590 in a series of calibrated $[Ca^{2+}]$-clamped solutions finely adjusted to include appropriate intracellular ingredients[13,14] (Methods). Fluorescence lifetime traces are normalised to their peak values (at ~1.3 ns post-pulse); the area under the curve over the time interval of 3 ns (tan shade) was measured and related to the peak value thus providing normalised total count (NTC) readout; $\lambda_x^{2p} = 910$ nm; temperature 33 °C. **b** The Cal-590 fluorescence lifetime imaging microscopy (FLIM) $[Ca^{2+}]$ sensitivity estimator, NTC, fitted with sigmoid-type function ($\chi^2 = 2.11 \times 10^{-5}$, $R^2 = 0.996$). **c** Top: Example of a CA3 pyramidal cell (SF-iGluSnFR channel); arrowheads, proximal part of the traced axon; patch pipette is seen. Bottom: Example of an axonal bouton (Cal-590 channel) traced from the cell soma; spiral line and arrow, tornado linescan applied in the middle of the bouton; scale bar, 50 μm (top) and 1 μm (bottom). **d** Example of a single-bouton Cal-590 signal during a 500-ms 100-Hz burst of spike-inducing somatic 1 ms current pulses: recorded as a tornado linescan (top; ordinate, spiral rotation angle 0–10π reflects five concentric spiral circles, 2π radian/360° each), fluorescence intensity (integrated over the 10π spiral scan) time course (middle), and fluorescence decay (FLIM, bottom; ordinate, decay time; grey arrowhead, laser pulse onset). Scale bars, 300 ms (top, horizontal), 2.0 ΔF/F (middle, vertical), 5 ns (bottom, vertical). **e** Intra-bouton Cal-590 fluorescence decay time course (normalised to peak) representing basal $[Ca^{2+}]$ (blue) and peak $[Ca^{2+}]$ (red), in the experiment shown in **d**

whether the A184S variant was suited to reliably report release of glutamate simultaneously from several boutons, we implemented a scanning mode in which the laser beam trajectory cycled across several boutons dwelling for 1–1.5 ms at each (Fig. 2a; Supplementary Figure 2b). The recorded data sets were arranged so that the fluorescence dynamics at individual boutons were shown as 'pseudo-linescans' with a 5–7-ms time step (Fig. 2b; Supplementary Figure 2c). The A184S indicator provided stable recordings over multiple trials and revealed no glutamate signal cross-talk between neighbouring boutons (Supplementary Figure 2d).

It turned out that these settings enabled not only direct readout of the average release probability $P_r$ (release success rate over a number of trials) but also an evaluation of the quantal properties (vesicular content) of release, trial-to-trial, at several synapses simultaneously. Quantal analyses were carried out by adapting a classical approach in which the signal amplitude histogram (SF-iGluSnFR.A184S ΔF/F readouts) was systematically fitted with multiple Gaussians, with the fitting parameters constrained by the noise (release-failure signal dispersion) and by the initial condition range of peak locations[21] from where it carried on unsupervised (Methods). The method revealed multiple, nearly equally spaced peaks in amplitude histograms, thus indicating the numbers and occurrences of released vesicles (up to three vesicular quanta per release; Fig. 2c, d; Supplementary Figure 2e-f;

this assessment can be further improved with larger trial numbers).

Finally, the combination of Cal-590 and SF-iGluSnFR.A184S enabled multiplex imaging of presynaptic $Ca^{2+}$ dynamics and glutamate release in similar settings (Supplementary Figure 3). However, in our hands the relatively slow kinetics of the A184S variant appeared suboptimal for imaging longer or higher-frequency AP trains. We therefore focussed on the combination of Cal-590 and SF-iGluSnFR.A184V fluorescence signals acquired in a fast-scanning multiplex tornado mode.

**Imaging quantal glutamate release and presynaptic $Ca^{2+}$.** First, we set out to monitor glutamate release and presynaptic $Ca^{2+}$ at axonal boutons scanned sequentially in pyramidal cells expressing SF-iGluSnFR.A184V and dialysed whole-cell with Cal-590 (Fig. 3a). To explore presynaptic function and its short-term plasticity, we recorded responses to four APs evoked (by the somatic pipette) at 20 Hz, a pattern of activation falling well within the range documented for CA3–CA1 connections in vivo[22]. Imaging was carried out in tornado mode, with the scanning spiral trajectory covering the visible bouton profile (Fig. 3b), as detailed previously[9]. Here the SF-iGluSnFR.A184V channel readily reported releases and failures in individual trials (Fig. 3b–d), whereas the Cal-590 channel showed reliable $Ca^{2+}$

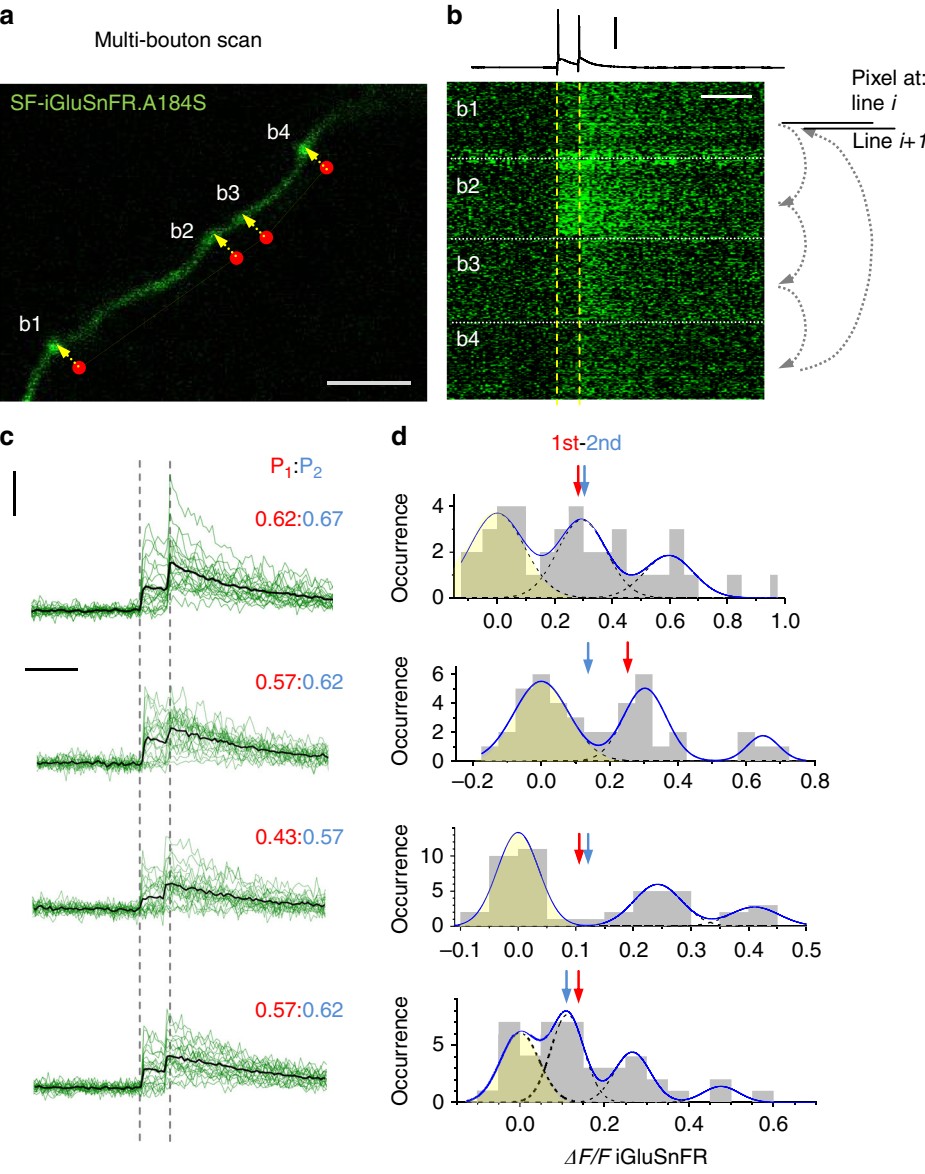

**Fig. 2** Multi-synapse imaging of quantal glutamate release with SF-iGluSnFR.A184S. **a** CA3 pyramidal cell axon fragment in area CA1 showing four presynaptic boutons (b1–b4); the scanning dwell points in the bouton centres (red dots, dwell-delay time ~1.5 ms per bouton) and laser scan trajectory (dotted yellow line) illustrated; scale bar, 5 μm. **b** A pseudo-linescan image of SF-iGluSnFR.A184S signals recorded simultaneously (one sweep example) at four boutons shown in **a** as indicated, during somatic generation of two action potentials 50 ms apart (top trace, current clamp). Arrow diagram relates displayed pixels to the scanning cycle: pixels at each displayed $i$th line are recorded sequentially among boutons (small arrows), and the next cycle fills the $(i+1)$th line of the display. Thus a pseudo-linescan image is generated showing brightness dynamics at individual boutons, with ~1.5 ms resolution; glutamate releases and failures can be seen; scale bars, 60 mV (vertical) and 100 ms (horizontal). **c** A summary of 22 trials in the experiment shown in **a**, **b**; green traces, single-sweep SF-iGluSnFR.A184S intensity readout at the four bouton centres; black traces, all-sweep-average; $P_1$:$P_2$, average probability $P_r$ (release success rate) of the first (red) and second (blue) release events; scale bars, 50% $\Delta F/F$ (vertical) and 100 ms (horizontal). **d** Amplitude histograms (SF-iGluSnFR.A184S $\Delta F/F$ signal, first and second response counts combined; prepulse baseline subtracted) with a semi-unconstrained multi-Gaussian fit (blue line, Methods) indicating peaks that correspond to estimated quantal amplitudes; the leftmost peak corresponds to zero signal (failure; yellow shade); dotted lines, individual Gaussians; arrows, average amplitudes (including failures) of the first (red) and second (blue) glutamate responses

entry signals (Fig. 3e), both channels displaying excellent signal-to-noise ratios. The cumulative photon count for Cal-590 appeared sufficient to monitor the average kinetics of pre-synaptic [Ca²⁺] (Fig. 3e, f; note that the Ca²⁺ fluorescence readouts reflect [Ca²⁺] values that are volume-equilibrated on the 2–5-ms scale).

In this one-bouton example (Fig. 3b), the SF-iGluSnFR.A184V $\Delta F/F$ signal histogram revealed quantal numbers and occurrences, suggesting up to four vesicles per release (Fig. 3g). The number of glutamate vesicles released at each individual trial reflects

(statistically) the 'instantaneous' vesicle probability $P_{r/v}$ at that trial, a quantity that may fluctuate or evolve in time depending on presynaptic conditions that might also fluctuate trial to trial[23–25]. The multiplex imaging thus enabled us to ask whether and how glutamate release depended on presynaptic Ca²⁺ homeostasis by examining statistical relationships between the two corresponding signals, trial by trial. We found that the amplitudes of glutamate-sensitive responses (AP-evoked $\Delta F/F$ increments, measured with the ~8 ms baseline subtracted) and the corresponding quantal content of release were positively correlated with the magnitude

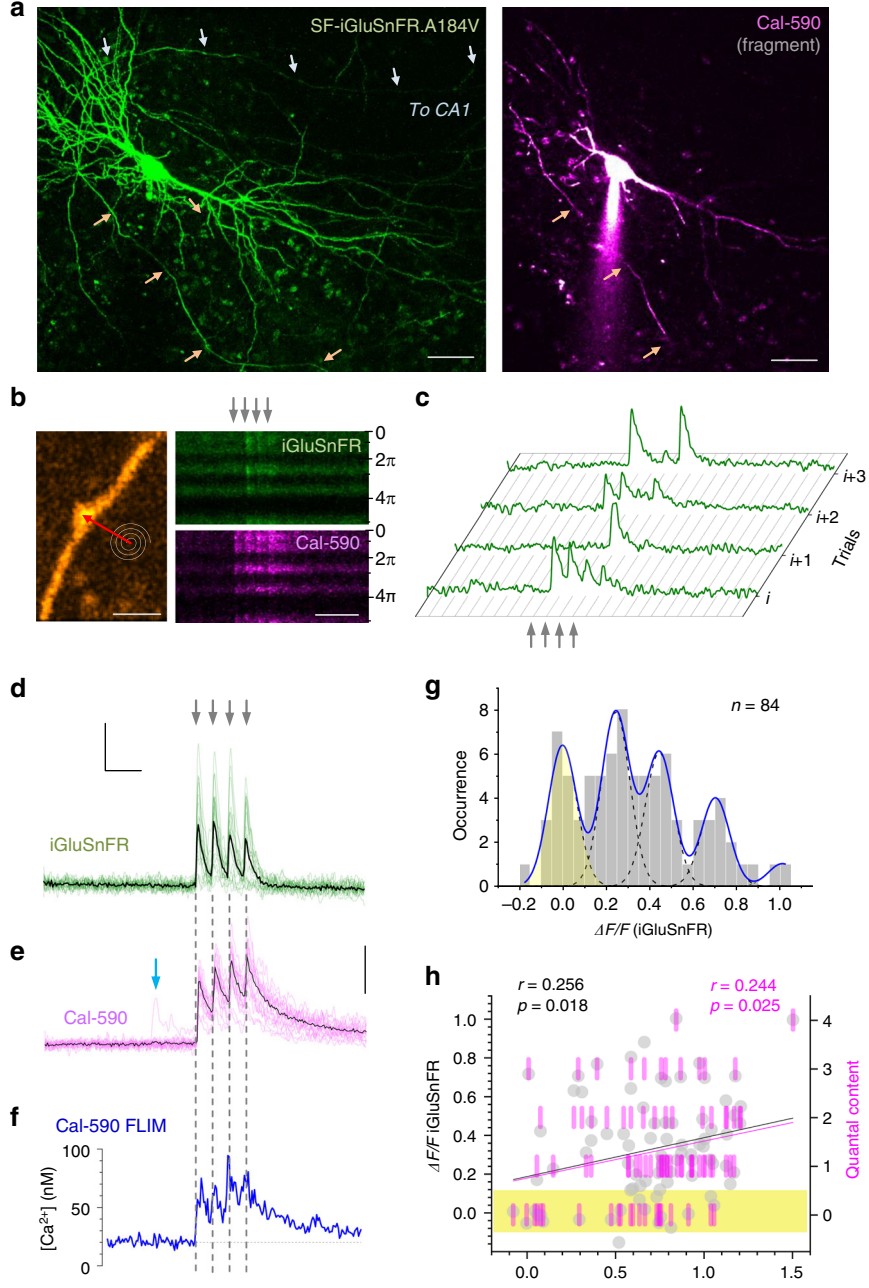

**Fig. 3** Multiplex imaging of quantal glutamate release and presynaptic Ca$^{2+}$ dynamics. **a** CA3 pyramidal cell. Left: SF-iGluSnFR.A184V channel[7] (Methods); arrows, two main axonal branches; ~60 μm z-stack, $\lambda_x^{2p} = 910$ nm. Right, Cal-590 channel (~20 μm z-stack fragment), whole-cell (300 μM Cal-590); patch pipette seen; scale bars, 20 μm. **b** Axonal bouton traced into CA1 from CA3 pyramid[9,49]: left, SF-iGluSnFR.A184V channel; tornado scan position shown; right, tornado linescan examples (rotation angle 0–5π rad reflects 2.5 spiral circles) during four evoked action potentials (APs; 20 Hz, arrows) shown in SF-iGluSnFR.A184V (green, note release failures) and Cal-590 (magenta) channels; scale bars, 2 μm (left), 200 ms (right). **c** Four characteristic sequential one-sweep recordings (SF-iGluSnFR channel) depicting individual quantal releases and failures. **d** Summary, glutamate release kinetics in bouton shown in **b** (SF-iGluSnFR.A184V channel; green, 20 trials 1 min apart; black, average); scale bars, 50% ΔF/F (vertical) and 100 ms (horizontal). **e** Summary, Cal-590 intensity signal ΔF/F (red channel) recorded as in **d** (magenta, individual trials; black, average); arrow, spontaneous Ca$^{2+}$ entry with no glutamate release (see **d**); scale bar, 30% ΔF/F. **f** Dynamics of free presynaptic [Ca$^{2+}$] averaged over 20 trials: Cal-590 fluorescence lifetime imaging microscopic readout (normalised photon count), converted to [Ca$^{2+}$]. Note: data reflect free ion concentration volume-equilibrated and time-averaged over 5–10 ms.
**g** Amplitude histograms (SF-iGluSnFR.A184V ΔF/F signal, first–to-fourth response counts combined; 8 ms pre-AP baseline subtracted); blue line, semi-constrained multi-Gaussian fit (Methods); peaks correspond to quantal amplitudes; leftmost peak, zero signal (release failure noise; yellow shade); false-positive cut-off at ~0.12 ΔF/F. **h** Trial-to-trial glutamate release signal amplitude (bouton in **b**) shown as ΔF/F SF-iGluSnFR.A184 readout (grey dots) and as quantal content (magenta bars, right ordinate; calculated from histogram in **g**) plotted against evoked Ca$^{2+}$ entry signal (ΔF/F Cal-590, as in **e**) for all recorded APs (8 ms pre-AP baseline subtracted). Black and magenta lines, linear regressions (r, Pearson's correlation; p, regression slope significance) for ΔF/F and quantal content data, respectively; yellow shade, failure response cut-off (as in **g**)

of concurrent AP-evoked [$Ca^{2+}$] increments (Cal-590 $\Delta F/F$; Fig. 3h). This correlation suggested that $P_{r/v}$ was dependent on the ongoing fluctuations in AP-evoked $Ca^{2+}$ entry.

One important consideration in the present context is that $Ca^{2+}$ indicators such as Cal-590, by buffering intracellular $Ca^{2+}$, could interfere with the endogenous $Ca^{2+}$ dynamics and thus neurotransmitter release properties (and possibly with the FLIM readout conditions). To gauge these potential concomitants, we first imaged a sub-set ($n = 7$ cells) of axonal boutons at relatively short times after whole-cell break-in, when Cal-590 had not yet equilibrated throughout the axon. Thus, during 20–22 recording trials (1 min apart), the axonal Cal-590 concentration, hence local $Ca^{2+}$ buffering capacity and its related effects, continued to rise, up to 2–3-fold, as clearly indicated by the rising total photon count of Cal-590 emission (Fig. 4a). Remarkably, the continued Cal-590 concentration rise had no effect on resting [$Ca^{2+}$] or the spike-evoked $Ca^{2+}$ entry measured using Cal-590 FLIM readout, which remained perfectly stable (Fig. 4b, c). This suggested no concomitant effects of Cal-590 fluctuations on FLIM readout, which was also in line with the tolerance of $Ca^{2+}$ homeostasis in neuronal processes to moderate levels of $Ca^{2+}$ buffering by OGB-1, as shown earlier[13]. Furthermore, when we compared axonal boutons between SF-iGluSnFR-expressing CA3 pyramidal cells loaded, or not loaded, with Cal-590, the average detected release probabilities $P_r$ were indistinguishable (Fig. 4d), lending further support to the physiological relevance of the method (see Discussion for detail).

The latter observation also indicated that $P_r$ varied ~20-fold among the recorded synapses (Fig. 4d). The multiple, poorly controlled aspects of synaptic organisation and function that underpin such heterogeneity in $P_r$ would mask any contributing effects of presynaptic $Ca^{2+}$ homeostasis. This well-known uncertainty underlines the importance of multiplexed monitoring of presynaptic $Ca^{2+}$ and glutamate release in individual synapses, in which case other concomitant contributing factors remain constant.

**Glutamate release depends on basal $Ca^{2+}$ and $Ca^{2+}$ entry**. To understand the relationship between presynaptic $Ca^{2+}$ and glutamate release further, we carried out multiplex imaging of Cal-590 and SF-iGluSnFR.A184V in multiple cell recording from multiple axonal boutons (Fig. 5a). The data revealed that the amount of released glutamate (SF-iGluSnFR.A184V $\Delta F/F$ readout) evoked by either a single AP or a four-AP train depended on presynaptic resting [$Ca^{2+}$] across the experiments (Supplementary Figure 4a-b). Reassuringly, this dependence was similarly strong when glutamate release was represented by its vesicular content (determined through quantal analyses; Fig. 5b, c). Similarly, the fluctuating amplitude of AP-evoked presynaptic [$Ca^{2+}$] entry was also linearly related to the amount of glutamate released (Fig. 5d; Supplementary Fig. 4c). These observations indicate that ongoing fluctuations in $P_{r/v}$ at small central synapses follow fluctuations in presynaptic [$Ca^{2+}$] dynamics, rather than occurring independently.

Intriguingly, neither resting [$Ca^{2+}$] nor the variability in evoked $Ca^{2+}$ entry had any detectable effect on the paired-pulse ratios or short-term plasticity indicators during the four-AP train (with 14/21 and 7/21 synapses showing, respectively, short-term facilitation and depression; Fig. 5e–g, Supplementary Figure 5).

**Nanoscopic geometry of $Ca^{2+}$ and glutamate imaging**. Neurotransmitter release is triggered by rapid $Ca^{2+}$ entry through presynaptic $Ca^{2+}$ channels. The effective distance between the prevalent $Ca^{2+}$ entry site and the release triggering site (SNARE protein complex) is considered the key to release probability

control[16,17,26]. What this distance is, whether it varies trial to trial (for instance, due to lateral channel migration or their stochastic opening), or whether it depends on mode of release remain debated[27].

To approach these questions, we first examined nanoscopic features of the present optical settings. Here axonal boutons express a membrane-bound (green) glutamate sensor and a cytosolic (red) $Ca^{2+}$ indicator (Fig. 6a). The inherent noise in the imaged bouton is because excitation and emission signals are blurred by the system's point-spread function (PSF), which is determined in large part by the optical diffraction limit. In a two-photon excitation microscope, the PSF is in the range of 0.2–0.4 μm in the (focal, $x$–$y$) plane of view and 0.8–1.5 μm in the $z$ direction. Mechanical fluctuations add to the blur. In these settings, a spiral (tornado) laser scan provides roughly uniform excitation across the bouton, with 0.2–0.4-μm resolution, in the focal plane, while integrating the signal in the $z$ direction: in other words, the imaged bouton reflects its projection on the $x$–$y$ plane (Fig. 6b).

Synaptic vesicle release generates a steep diffusion gradient of glutamate, with its extracellular concentration dropping orders of magnitudes over tens of nanometres from the release site[16,17,26,28,29]. Thus its fluorescent signal generated by SF-iGluSnFR.A184V is expected to show a nanoscopic hotspot with a spread reflecting glutamate escape. However, because this signal originates from the three-dimensional bouton it is important to understand how it projects onto the $x$–$y$ plane. The bouton projection that is registered by the circular tornado scan (Fig. 6c) underestimates curvilinear (geodesic) distances on the bouton surface. Measuring the profiles of 26 recorded axonal boutons suggested that most could be reasonably approximated by rotational ellipsoids (Supplementary Figure 6a-b), with the sample-aevarge rotation radius of $0.89 \pm 0.04$ μm (mean ± s.e.m.) and the major axis radius (along the axon trajectory) of $1.18 \pm 0.04$ μm (mean ± s.e.m.; Supplementary Figure 6c).

Geodesic correction for curvilinear distances on the spherical surface of radius $b$ is straightforward: the projected distance $x_i$ from the profile centre will correspond to the curvilinear geodesic distance of $b \arcsin (x_i/b)$ (Fig. 6d, left). Obtaining the exact correction for an elliptical surface (Fig. 6d, right) could be much more complicated. However, in our experimental sample the average ratio of major and minor ellipsoid axes $a/b$ for axonal boutons is $1.35 \pm 0.04$ (mean ± s.e.m.; Supplementary Figure 6c). In the conservative case of $a/b = 1.5$, one can show that the difference between the corrected distances for the ellipsoidal surface (radii $a$ and $b$) and the spherical surface (radius $a$) is unlikely to exceed ~5% (Supplementary Figure 6e-f). Thus the visible planar map of the ellipsoidal bouton imaged with a circular flat scan (SF-iGluSnFR.A184V $\Delta F/F$ signal in Fig. 6e–f) will stretch the visible lengths in the directions $x$ and $y$ by factors $b \arcsin (x_i/b)$ and $a \arcsin (y_i/a)$, respectively, to form a corrected geodesic surface map (Fig. 6g).

In addition to the geodesic distance correction, fluorescence intensity signal collected in a planar projection is overestimated towards the edge of spherical or ellipsoidal shapes (Supplementary Figure 6g, inset). In the present study, we focused on the 'ratiometric' $\Delta F/F$ SF-iGluSnFR.A184V signal, which should effectively nullify this bias. However, because this correction could prove useful in other cases, we have detailed its theory and validated it using a micro-vesicle with a fluorescent shell (Supplementary Figure 6h-i).

**Localising glutamate release and $Ca^{2+}$ entry on the nanoscale**. In 23 analysed presynaptic boutons, tornado scan heat maps (averaged over 20–22 consecutive trials, 1 min apart, in each

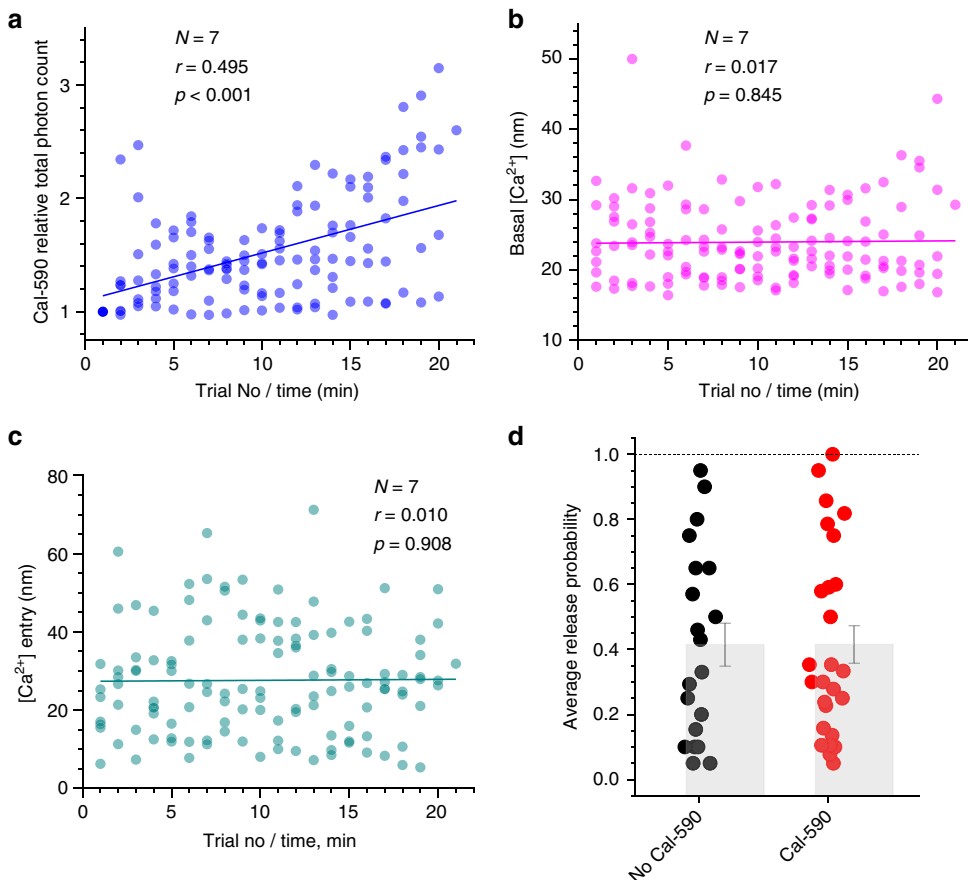

**Fig. 4** Intra-axonal $Ca^{2+}$ buffering by Cal-590 has little effect on $[Ca^{2+}]$ fluorescence lifetime imaging microscopic (FLIM) readout or average release probability. **a** Total photon count for Cal-590 (fluorescence intensity, measured within 300 ms before the evoked four-action potential (four-AP) train at 20 Hz, as in Figs. 3 and 4), normalised to the initial value, as recorded in individual axonal boutons ($N = 7$), plotted against the trial number (time). In these experiments, the intra-axonal Cal-590 concentration, hence $Ca^{2+}$ buffering capacity, continues to rise with time, before eventual equilibration; solid line, linear regression ($r$, Pearson correlation; $p < 0.001$, slope significance). **b** Presynaptic resting $[Ca^{2+}]$ measured within 300 ms before the evoked four-AP train using Cal-590 FLIM readout, at axonal boutons shown in **a**, plotted against the trial number (time); other notation as in **a** (no correlation). **c** An increment in presynaptic $[Ca^{2+}]$ during evoked four-AP train measured with Cal-590 FLIM, at axonal boutons shown in **a**, **b**, plotted against the trial number (time); other notation as in **a**, **b** (no correlation). **d** Probability of evoked glutamate release (in response to a single action potential) in axonal boutons of CA3 pyramidal cells (organotypic hippocampal slices) expressing SF-iGluSnFR, with and without Cal-590 (300 μM) being loaded and equilibrated in whole-cell mode, as indicated. Dots, individual bouton recordings; bar graph, mean ± s.e.m. (0.41 ± 0.07 and 0.42 ± 0.06, $n = 20$ and $n = 26$, without and with Cal-590, respectively)

bouton) revealed clear hotspots for the *ΔF/F* SF-iGluSnFR.A184V signal (1 peak in 20/23 and 2 separate peaks in 3/23 synapses, Fig. 7a), with a >10-fold drop in the *ΔF/F* signal intensity profile within 1–1.5 μm from the peak (Fig. 7b). The fact that the signal averaging over multiple trials reveals individual hotspots suggests that trial-by-trial glutamate release is constrained to a relatively narrow area of the bouton membrane. We next used geodesic correction to establish the spatial spread of the glutamate signal across the bouton surface. The average intensity profile normalised to the peak value was well fitted by a single exponent with a spatial decay constant of 0.547 ± 0.016 μm (mean ± s.e.m.; $n = 23$ boutons; Fig. 7b). This was in agreement with biophysical studies predicting little glutamate escape beyond 0.5–1.0 μm at hippocampal synapses[30], mainly due to the powerful, high-affinity uptake by the surrounding astroglia[31].

In these experiments, multiplex imaging of Cal-590 and SF-iGluSnFR.A184V provided the opportunity to assess spatial colocalisation of $Ca^{2+}$ entry and the glutamate release site (Fig. 7c; Methods). While the glutamate release SF-iGluSnFR.A184V *ΔF/F* heat maps consistently displayed well-identified hotspots (Fig. 7a±c), the Cal-590 FLIM readout signal averaged over 20–22 trials in individual boutons showed much lower signal-to-noise ratios. This was likely due to a combination of factors including a low photon count, trial-to-trial variability in the pattern of activated $Ca^{2+}$ channels, their lateral trial-to-trial movement, or even the rapid diffuse spread of Ca-bound Cal-590[27] (see Discussion). Nonetheless, $Ca^{2+}$ entry hotspots (>2SD above the noisy background) were detected in at least nine synaptic boutons (Fig. 7c; because $Ca^{2+}$ hotspots could occur within the bouton volume, these maps were not corrected for geodesic distances). These boutons provided data to assess whether the juxtaposition pattern for $Ca^{2+}$ entry and glutamate release occurred purely by chance. To this end, we ran a straightforward statistical test in which the experimental scatter of distances between $Ca^{2+}$ and glutamate hotspots was compared with the scatter of distances among arbitrary pairs of points randomly scattered over a similar circular area (Fig. 7d). The test revealed that the experimental distances were significantly lower than the simulated ones (Fig. 7e), although only 4 out of 9 boutons showed this distance in the 0–200 nm range (boutons 6–9 in Fig. 7c).

The results suggested that $Ca^{2+}$ entry occurs significantly closer to the site of glutamate release than it would be expected purely by chance. However, this observation could reflect a simple

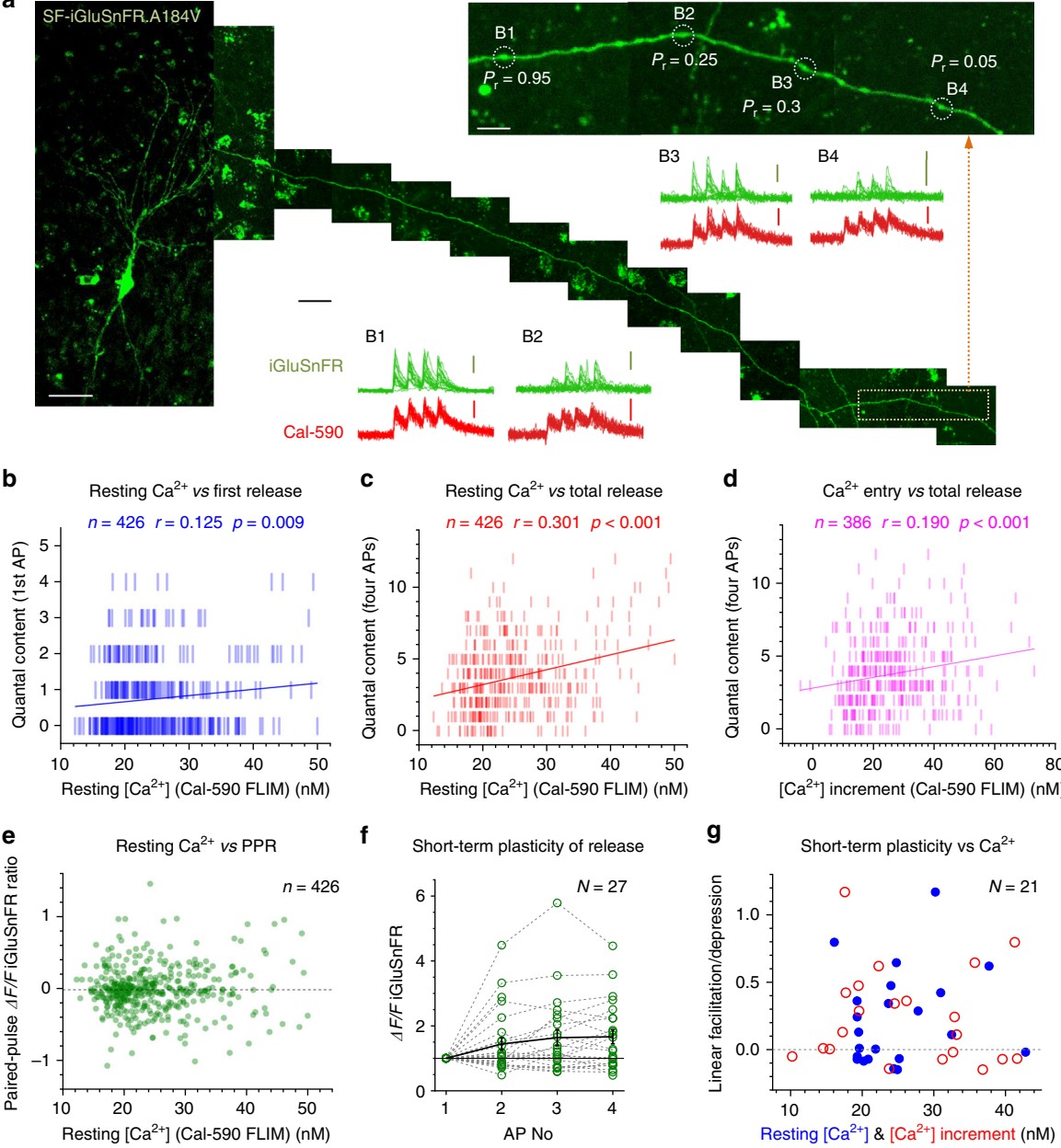

**Fig. 5** Resting presynaptic Ca$^{2+}$ and evoked Ca$^{2+}$ entry control glutamate release but not its short-term plasticity. **a** Collage, CA3 pyramidal cell with axon traced into area CA1 (example; SF-iGluSnFR channel; $xy$ projections of 10–15 μm $z$-stacks); inset, zoomed-in axonal fragment (orange dotted rectangle) depicting four axonal boutons B1–B4 ($P_r$, average release probability); scanning mode: tornado. Traces, summary recordings of glutamate release (green) and Ca$^{2+}$ dynamics (red), as indicated; scale bars, 20 μm (left image), 100 ms (horizontal middle), 50% $\Delta F/F$ (green, SF-iGluSnFR.A184V), and 15% $\Delta F/F$ (red, Cal-590), 5 μm (right image). **b** Quantal content of glutamate release upon first action potential (AP), plotted against resting [Ca$^{2+}$] (Cal-590 fluorescence lifetime imaging microscopic (FLIM) readout, averaged over 100 ms prepulse). Blue bars, the number of released vesicles estimated from the frequency histogram of the SF-iGluSnFR.A184V $\Delta F/F$ signal (as in Fig. 3g, h). Solid blue line, linear regression ($r$, Pearson's correlation; $p$, regression slope significance; $n$, number of events, $N = 26$ boutons recorded). See Supplementary Figure 4a for raw $\Delta F/F$ data. **c** Quantal content of cumulative glutamate release upon four APs, plotted against presynaptic resting [Ca$^{2+}$] (Cal-590 FLIM readout); other notations as in **b**. See Supplementary Figure 4b for raw $\Delta F/F$ data. **d** Cumulative glutamate release upon four APs (as in **c**; tests with two APs excluded), plotted against cumulative [Ca$^{2+}$] increment (Cal-590 FLIM readout during four APs); other notations as in **b** ($N = 24$ boutons). See Supplementary Figure 4c for raw $\Delta F/F$ data. **e** Paired-pulse ratio (PPR, ratio between second and first $\Delta F/F$ SF-iGluSnFR signal amplitudes, pre-AP 8 ms baselines subtracted) plotted against resting [Ca$^{2+}$]. Other notations as in **b**. **f** Amplitude of $\Delta F/F$ SF-iGluSnFR signal evoked by 1–4 APs, relative to the first-AP $\Delta F/F$ signal, plotted against the AP number (1–4; $N$, number of recorded boutons). **g** Short-term facilitation/depression of AP burst-evoked glutamate release (linear regression slope over 1–4 $\Delta F/F$ SF-iGluSnFR signal change shown in **f**; tests with two APs were excluded) plotted against resting [Ca$^{2+}$] (blue dots) and AP-evoked [Ca$^{2+}$] increment (red circles). See Supplementary Figure 5 for trial-to-trial short-term plasticity readout, plotted against [Ca$^{2+}$] parameters

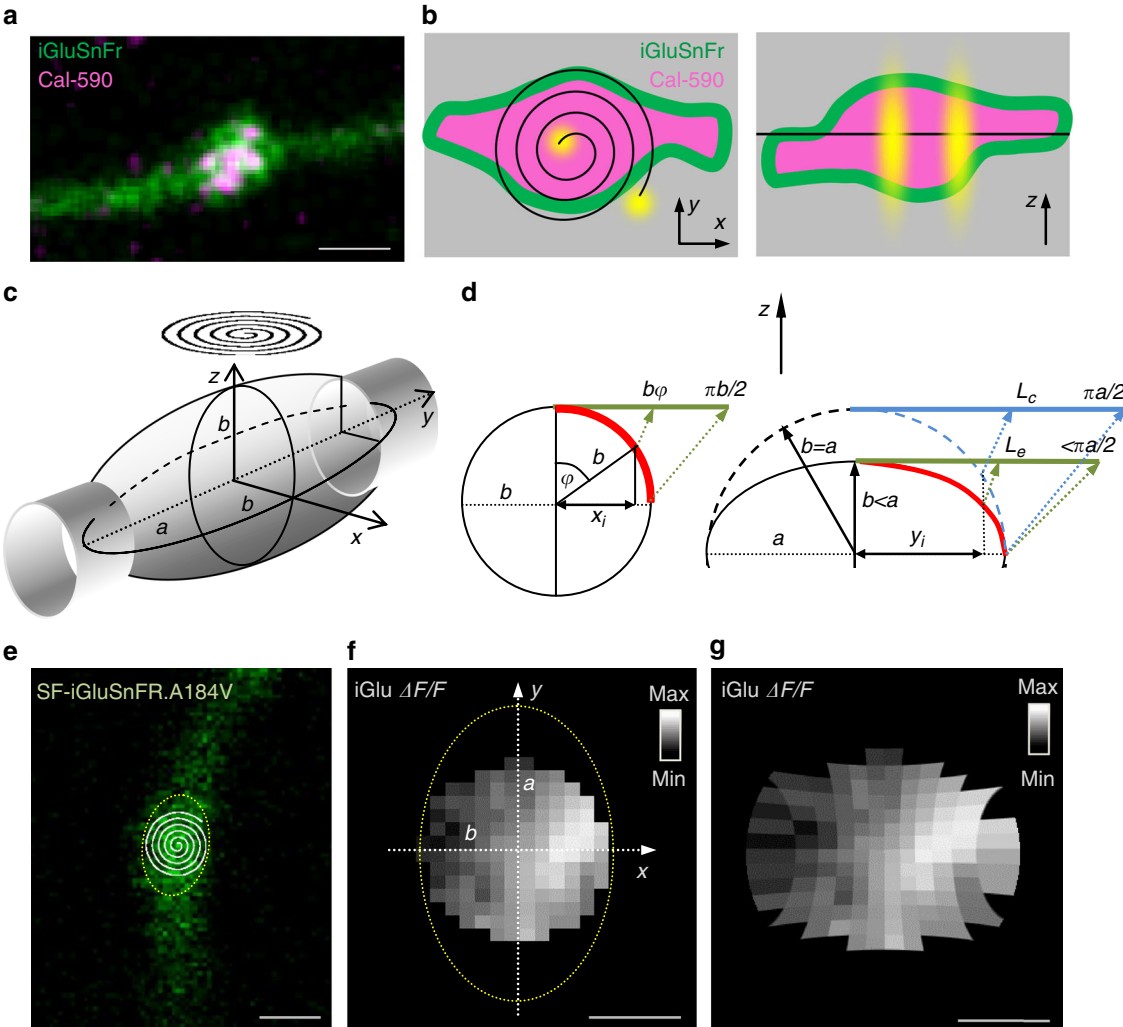

**Fig. 6** Nanoscale geometry of presynaptic bouton imaging with two-photon excitation. **a** CA3–CA1 axonal bouton (example), two-channel image (SF-iGluSnFR.A184V green, Cal-590 magenta); scale bar, 1 μm. **b** Axonal bouton diagram illustrating laser scanning settings, x–y (left) and z (right) planes, with membrane-bound SF-iGluSnFR (green) and cytosolic Cal-590 (magenta); black spiral (left) and straight line (right), tornado linescan trajectory; yellow shapes, characteristic point-spread function (PSF). **c** Diagram, typical experimental arrangement, with a tornado linescan (spiral) over an ellipsoidal axonal bouton (a and b, major and minor axes, respectively, in the elliptical x–y projection of the rotational ellipsoid). **d** Trigonometric diagrams explaining geodesic corrections for curved-surface distances projected onto the x–y plane. Left, for a circle or cylinder of radius b, the projected distance $x_i$ from the centre corresponds to the curvilinear (geodesic) distance $b\varphi$ where angle $\varphi = \arcsin(x_i/b)$ (green segment, geodesic distance 'straightened' in plane of view); thus the projected distance from the centre to the edge, b, corresponds to the geodesic distance $\pi b/2$. Right, for an elliptical section with major and minor axes a and b, respectively, the projected distance $x_i$ from the centre corresponds to the geodesic distance ($L_e$, green segment), which is smaller than that for a circular correction ($L_c$, blue segment). The difference between $L_e$ and $L_c$ depends on the expected a/b ratio (see Supplementary Figure 4d–f for detail). **e** Example of a recorded bouton (SF-iGluSnFR.A184V channel), with a tornado scan shown; dotted oval, an estimated outline of the axonal bouton projection; scale bar, 1 μm. **f** Characteristic heat map of the ΔF/F SF-iGluSnFR.A184V fluorescence signal generated by action potential-evoked glutamate release (one-bouton example; imaging arrangement as in **e**, circular shape follows the tornado scan); average of 21 trials 1 min apart; dotted oval, bouton outline as shown in **e**; image projected on to the focal x–y plane; scale bar, 0.5 μm. **g** Heat map of ΔF/F SF-iGluSnFR as in **f** corrected for geodesic distances as opposed to the projected (visible) distances, in accord with the geometry of the bouton outline and the tornado scan position; scale bar, 0.5 μm

fact that presynaptic $Ca^{2+}$ channels do group towards release sites, rather than being scattered randomly around the bouton. More importantly, we found no nanoscopic co-localisation of $Ca^{2+}$ entry and glutamate release suggesting no tight coupling, as discussed below.

## Discussion

In this study, we have established that the fluorescence lifetime of the red-shifted $Ca^{2+}$ indicator Cal-590 is sensitive to $[Ca^{2+}]$ in the 10–200 nM range. We have calibrated this dependence using the normalised photon counting protocol described earlier[13,14]

and thus been able to obtain readout of $[Ca^{2+}]$ dynamics in neuronal axons using whole-cell dialysis with Cal-590 in organotypic brain slice preparations. The knowledge about low presynaptic $[Ca^{2+}]$ is fundamentally important for understanding the machinery of neurotransmitter release. Presynaptic $[Ca^{2+}]$ is controlled by active $Ca^{2+}$ sensors, such as $Ca^{2+}$-sensitive channels and pumps, and is in a dynamic equilibrium with local $Ca^{2+}$ buffers. Thus changes in basal $[Ca^{2+}]$ not only report a shifted equilibrium but also disproportionately alter the availability of $Ca^{2+}$-free buffers[32–34] as the bound-to-free ratio for presynaptic $Ca^{2+}$ varies between 100- and 1000-fold[19,26,35]. The cytosolic $Ca^{2+}$ buffering capacity affects the spatiotemporal

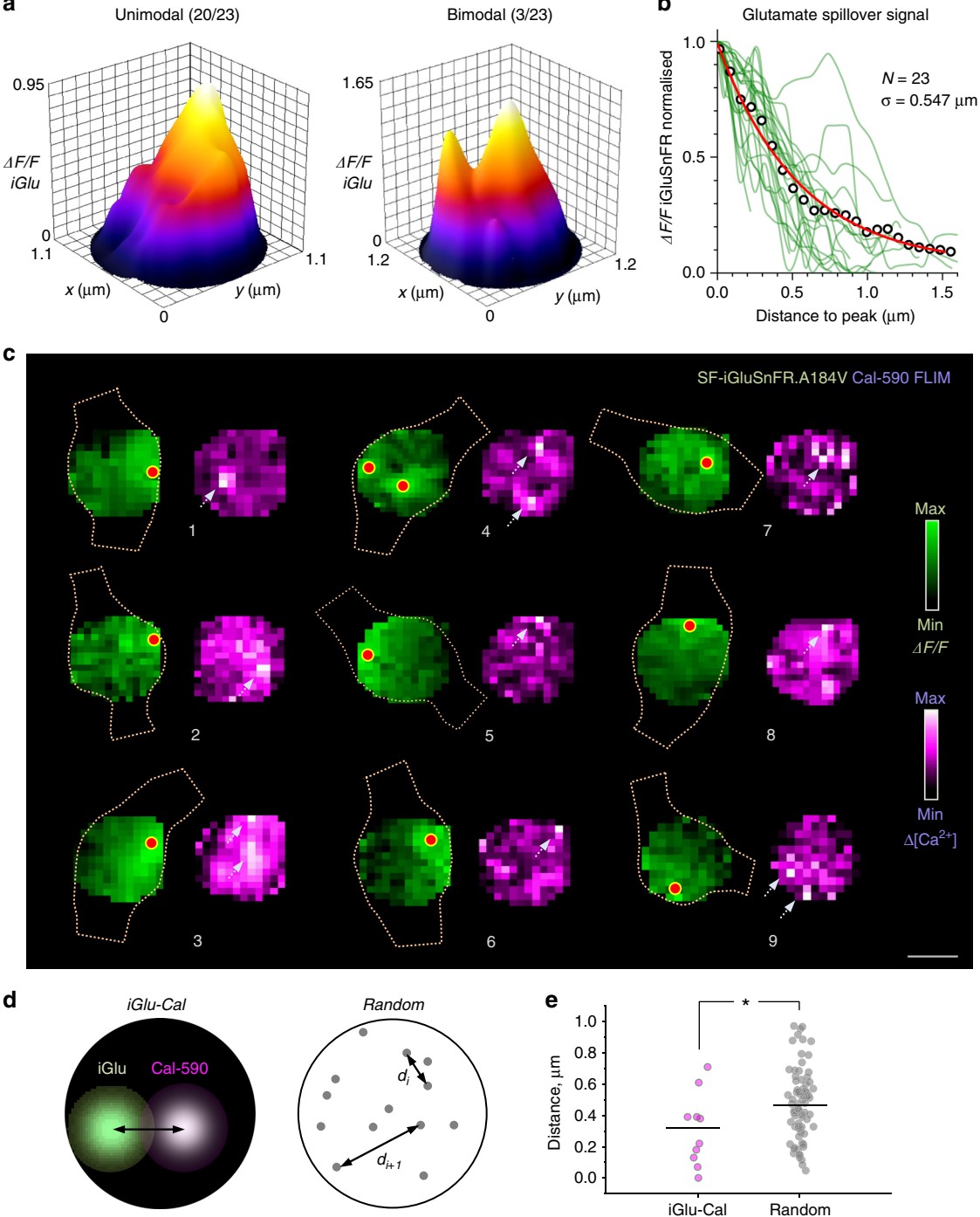

**Fig. 7** Evaluating sub-microscopic glutamate signal spread and co-localisation of glutamate release and presynaptic $Ca^{2+}$ entry in presynaptic boutons. **a** Examples of unimodal (left, 20/23 boutons) and bi-modal (right, 3/23 boutons) spatial profiles of the SF-iGluSnFR signal intensity (20-22 trial average) upon glutamate release, as seen in the focal plane; averaging spatial filtering (~100 nm range) applied; peaks point to the most likely release site location. **b** Spatial decay of the fluorescence signal ($\Delta F/F$ SF-iGluSnFR.A184V) measured in geodesic-corrected heat maps (as in Fig. 6g) for $N = 23$ individual boutons (green lines); circles, average; solid red line, best-fit exponent: spatial decay constant $\sigma = 0.547 \pm 0.016\ \mu m$ (mean ± standard error). **c** Examples of simultaneously recorded heat maps showing glutamate signal (green, $\Delta F/F$ SF-iGluSnFR) and $[Ca^{2+}]$ increment profiles (magenta, Cal-590 fluorescence lifetime imaging microscopic readout upon four action potentials, magenta); dotted shapes, outlines of the recorded axonal bouton profiles; red dots, estimated glutamate release site locations; arrows, consistent hotspots of Cal-590 $[Ca^{2+}]$ transients; scale bar, 0.5 μm. **d** Diagrams illustrating the measurement of distances between recorded hotspots of glutamate and $Ca^{2+}$ signals, as shown in **c** (left), and between points randomly scattered within a similar circular area (right). **e** The average distance between recorded hotspots of glutamate and $Ca^{2+}$ signals (mean ± s.e.m., 0.308 ± 0.073 μm, $n = 10$; grey dots) is significantly lower than that between randomly scattered points (0.474 ± 0.025, $n = 80$, orange dots; *$p < 0.028$, $t$-test)

profile of $Ca^{2+}$ entry, hence the triggering of release[16–18]. The range of presynaptic $[Ca^{2+}]$ values we have detected is remarkably low, representing only a few free $Ca^{2+}$ ions per cubic micron; however, these are never the same few molecules, as bound and free $Ca^{2+}$ are in a continued rapid exchange.

In addition to high $[Ca^{2+}]$ sensitivity in the nanomolar range (for OGB-1 or Cal-590), a critical advantage of FLIM readout is that it does not depend on the dye amount or concentration, light scattering, focus fluctuations, and other concomitants of live imaging[13,14]. The latter is particularly important for imaging in awake or freely moving animals. Having a relatively more complicated procedure to record and analyse FLIM recordings compared to standard fluorometric monitoring[13,14] appears a small price to pay for such advantages.

Because $Ca^{2+}$ indicators are, by definition, $Ca^{2+}$ buffers, they could affect intracellular $Ca^{2+}$ homeostasis. However, moderate amounts of intracellular OGB-1 had little influence on the cell-wide landscape of basal $Ca^{2+}$ in principal neurons[13]. This is because equilibrated intracellular $[Ca^{2+}]$ levels are controlled by active $[Ca^{2+}]$-sensing mechanisms rather than by passive $Ca^{2+}$ buffers[33]. At the same time, the effect of $Ca^{2+}$ indicators on rapid $Ca^{2+}$ dynamics and release probability (both average $P_r$ and instantaneous $P_{r/v}$) could be significant, as we found previously with OGB-1[19,36]. Nonetheless, Cal-590 has a relatively low affinity ($K_d \sim 560\,nM$) compared to OGB-1 ($K_d \sim 170–260\,nM$)[12,37], and therefore should have a lower $Ca^{2+}$ buffering effect. Indeed, we found no detectable influence of Cal-590 on resting presynaptic $[Ca^{2+}]$, spike-evoked (bouton volume-equilibrated) $Ca^{2+}$ entry, or the average $P_r$ (Fig. 4). It appears therefore that the changes in rapid presynaptic $Ca^{2+}$ dynamics due to the presence of Cal-590 (in the used range) are too small to influence neurotransmitter release. While this speaks in favour of the method, the physiological implications of this observation require a separate study: one plausible scenario is that, with or without added $Ca^{2+}$ indicators, presynaptic $Ca^{2+}$ transients 'saturate' the release machinery on its $Ca^{2+}$-sensing side. Reassuringly, it was earlier reported that bolus-loading application of Cal-590 was fully compatible with intense network spiking activity in the visual cortex in vivo[15]. Furthermore, $Ca^{2+}$ indicators with higher affinity, such as OGB-1 or some genetically encoded $Ca^{2+}$ indicators, have been widely used to monitor neural network activity in freely moving animals, without apparent behavioural implications[32,38]. A systematic examination of axons loaded with Cal-590 at different concentrations is required to quantitatively assess potential subtle effects of the dye.

Simultaneous readout of presynaptic $Ca^{2+}$ dynamics and glutamate release at small central synapses in situ is important for at least two reasons. First, glutamate signal readout provides monitoring of stochastic release (and thus release probability) under varied physiological conditions. At the same time, axonal $Ca^{2+}$ monitoring is a reliable indicator of presynaptic spikes: the efficiency of (bolus-applied) Cal-590 in registering APs in deep brain imaging settings has recently been demonstrated[15]. Thus, comparing the occurrence of spikes in $Ca^{2+}$ channel with the statistics of stochastic glutamate release events in the SF-iGluSnFR channel at axonal boutons should provide a quantitative gauge of synaptic transmission fidelity.

Second, the method provides insights into the mechanistic, albeit statistical, relationship between $Ca^{2+}$ entry and vesicle release probability $P_{r/v}$ and how this relationship changes during use-dependent synaptic plasticity in individual synapses. To address such fundamental questions, classical studies in giant (Calyx) synapse preparations combined presynaptic $Ca^{2+}$ imaging with patch-clamp recordings in presynaptic and/or postsynaptic structures, which were large enough to allow access[10,11,39]. By being able to probe and manipulate presynaptic

mechanisms in individual synapses and thus to avoid multiple concomitant factors contributing to inter-synaptic variability of release, such studies have been pivotal for understanding the basics of $Ca^{2+}$-dependent release machinery. Our multiplexed imaging method enables a similar type of probing in common small synapses.

The present study reveals that fluctuations in basal presynaptic $[Ca^{2+}]$ as well as in the AP-evoked $Ca^{2+}$ entry affect release efficacy. The role of presynaptic depolarisation-induced rises in basal $[Ca^{2+}]$ for neurotransmitter release has previously been shown in giant calyceal synapses[40]. Our findings suggest that the relationship between basal $[Ca^{2+}]$ and release efficacy is maintained in small central synapses, in quiescent tissue conditions in situ. Whether the observed fluctuations in presynaptic resting $[Ca^{2+}]$ occur purely stochastically or are driven by the external (network) factors remains to be ascertained. Similarly, the observation that fluctuations in AP-evoked $Ca^{2+}$ entry are also correlated with glutamate release makes such fluctuations a significant contributor to release variability, in addition to other presynaptic mechanisms.

The kinetics of the glutamate-sensor SF-iGluSnFR.A184S and that of Cal-590 appear well suited to monitor release probability (both $P_r$ and $P_{r/v}$) at multiple synapses supplied by the same axon. This enables real-time readout of heterogeneous presynaptic identities, their possible cooperative features, and their plasticity within an identified synaptic circuit. On a technical note, here we used the basal SF-iGluSnFR fluorescence signal, under sparse expression of the indicator, to trace individual cell axons. Clearly, this axonal tracing approach could be adapted to the use of other appropriate dyes and genetically encoded indicators (such as a red-shifted version of iGluSnFR[41] or sensors for other neurotransmitters) or other sparse transfection methods to enable simultaneous multi-target in vivo functional imaging. In addition to transfection of indicators, co-transfection with RNAi or CRISPR constructs should offer a method to assess the molecular signalling mechanisms underlying glutamate release and short-term plasticity[42].

Release of glutamate involves membrane fusion of a 40-nm synaptic vesicle, generating a sharp neurotransmitter diffusion gradient inside and outside the cleft[29,43]. Similarly, $Ca^{2+}$ entering presynaptic boutons through a cluster of voltage-gated $Ca^{2+}$ channels dissipates orders of magnitude at nanoscopic distances from the channel mouth[16,17,26]. Thus, in theory, in both cases we should detect a nanoscopic signal hotspot. In recorded images, such hotspots will be blurred by the microscope's PSF and the inherent experimental noise. However, taking advantage of averaging under multiple exposure over repeated trials, we were able to localise the preferred regions of glutamate release and, in some cases, $Ca^{2+}$ entry, beyond the diffraction limit in the $x$–$y$ plane (Fig. 7a–c). Clearly, such accuracy in the $z$ direction appears more problematic: it will require specific optics that would enable registration of optical aberration (such as astigmatism) against the $z$ coordinate[44] or otherwise a specifically modified PSF shape[45].

In our experiments, all tested synapses displayed a clear nanoscopic hotspot of glutamate release (in average SF-iGluSnFR signal heat maps), suggesting consistent localisation of the glutamate release site from trial to trial (Fig. 7a, b). This appears in line with the notion that some key presynaptic proteins within the active zones (in particular, Rab3-interacting molecule, associated with release machinery) closely align, across the synaptic cleft, with postsynaptic receptors, in a trans-synaptic molecular 'nanocolumn'[46]: such alignment would appear functionally irrelevant with a variable release site. Glutamate release localisation could also be achieved by imaging vesicle-associated pHluorins that increase fluorescence intensity upon vesicle fusion[5]. An

elegant variant of this method, termed pHuse, has used nano-localisation of individual release events in cultured cells to detect spatially constrained glutamate release sites at least in a group of relatively small presynaptic boutons[46], in line with the present observations.

In contrast, we found that presynaptic sites of $Ca^{2+}$ entry were largely scattered over the bouton face during multiple trials, with only nine boutons showing discernible peaks (Fig. 7c). Among these nine synapses, only some displayed co-localisation of $Ca^{2+}$ and glutamate signal hotspots on the small scale (<200 nm). Whilst the rapid diffuse spread of a soluble Ca-bound indicator (unlike the membrane-bound SF-iGluSnFR indicator) could blur its nanoscale fluorescence intensity profile post-AP[20,27,36], the FLIM readout should not generally depend on the indicator concentration. The apparent lack of tight co-localisation lends support to the hypothesis of loose coupling between presynaptic $Ca^{2+}$ channels and synaptic release machinery, which seems characteristic for plastic synapses[17,18] and has been suggested by biophysical models of axonal boutons at CA3–CA1 connections[47]. The loose-coupling hypothesis would also appear consistent with the finding that trial-to-trial fluctuations in presynaptic basal $Ca^{2+}$ or $Ca^{2+}$ entry affect glutamate release efficacy (Fig. 5b–d): in such cases, release should depend on the variable cooperative action of multiple $Ca^{2+}$ channels rather than on one channel opening[16]. Furthermore, as a large proportion of recorded boutons did not show clear $Ca^{2+}$ entry hotspots in the all-trial-average heat maps, the site of $Ca^{2+}$ entry might not be consistent from trial to trial, either because of stochastic opening among a set of presynaptic $Ca^{2+}$ channels, due to their lateral motility[48], or indeed because of the fluctuating contribution from local $Ca^{2+}$ stores[49].

## Methods

### Animal experimentation.
All experiments involving animals were carried out in accordance with the European Commission Directive (86/609/EEC) and the United Kingdom Home Office (Scientific Procedures) Act (1986) under the Home Office Project Licence PPL P2E0141 E1.

### Organotypic slice culture preparation.
Organotypic hippocampal slice cultures were prepared and grown with modifications to the interface culture method[50] from P6–8 Sprague-Dawley rats. Three hundred-μm-thick, isolated hippocampal brain slices were sectioned using a Leica VT1200S vibratome in ice-cold sterile slicing solution consisting (in mM) of Sucrose 105, NaCl 50, KCl 2.5, $NaH_2PO_4$ 1.25, $MgCl_2$ 7, $CaCl_2$ 0.5, Ascorbic acid 1.3, Sodium pyruvate 3, $NaHCO_3$ 26, and Glucose 10. Following washes in culture media consisting of 50% Minimal Essential Media, 25% Horse Serum, 25% Hanks Balanced Salt solution, 0.5% L-Glutamine, 28 mM Glucose, and the antibiotics penicillin (100 U/ml) and strep-tomycin (100 μg/ml), three to four slices were transferred onto each 0.4-μm pore membrane insert (Millicell-CM, Millipore, UK), kept at 37 °C in 5% $CO_2$ and fed by medium exchange every 2–3 days for a maximum of 21 days in vitro (DIV). The slices were transferred to a microscope recording chamber with the recording artificial cerebrospinal fluid solution containing (in mM): NaCl 125, $NaHCO_3$ 26, KCl 2.5, $NaH_2PO_4$ 1.25, $MgSO_4$ 1.3, $CaCl_2$ 2, and glucose 16 (osmolarity 300–305 mOsm), continuously bubbled with 95% $O_2$/5% $CO_2$. Recordings were carried out at 33–35 °C, with addition of 10 μM NBQX and 50 μM AP5 to reduce the potential for plasticity effects influencing synaptic properties during prolonged recordings.

### Biolistic transfection of SF-iGluSnFR variants.
Second-generation iGluSnFR variants SF-iGluSnFR.A184S and SF-iGluSnFR.A184V[7] were expressed under a synapsin promoter in CA3 pyramidal cells in organotypic slice cultures using biolistic transfection techniques adapted from the manufacturer's instructions. In brief, 6.25 mg of 1.6 micron Gold micro-carriers were coated with 30 μg of SF-iGluSnFR plasmid. Organotypic slice cultures at 5 DIV were treated with culture media containing 5 μM Ara-C overnight to reduce glial reaction following transfection. The next day, cultures were shot using the Helios gene-gun system (Bio-Rad) at 120 psi. The slices were then returned to standard culture media the next day and remained for 5–10 days before experiments were carried out.

### Axon tracing and imaging of presynaptic boutons.
We used a Femtonics Femto2D-FLIM imaging system integrated with patch-clamp electrophysiology (Femtonics, Budapest) and linked on the same light path to two femtosecond pulse lasers MaiTai (SpectraPhysics-Newport) with independent shutter and intensity control. Patch pipettes were prepared with thin-walled borosilicate glass capillaries (GC150-TF, Harvard apparatus) with open tip resistances 2.5–3.5 MΩ. For CA3 pyramidal cells, internal solution contained (in mM) 135 potassium methane-sulfonate, 10 HEPES, 10 di-Tris-Phosphocreatine, 4 $MgCl_2$, 4 $Na_2$-ATP, and 0.4 Na-GTP (pH adjusted to 7.2 using KOH, osmolarity 290–295) and supplemented with Cal-590 (300 μM; AAT Bioquest) for FLIM imaging.

Presynaptic imaging was carried out using an adaptation of previously described methods for separate presynaptic glutamate and $Ca^{2+}$ imaging[9]. Cells were first identified as SF-iGluSnFR expressing using two-photon imaging at 910 nm and patched in whole-cell mode as above. Following break-in, 30–45 min were normally allowed for Cal-590 to equilibrate across the axonal arbour; in a sub-set of experiments, shorter post-break-in times were implemented to test release properties under the continued equilibration (concentration rise) of Cal-590 inside the axon. Axons, identified by their smooth morphology and often tortuous trajectory, were followed in frame scan mode to their targets and discrete boutons were identified by criteria previously demonstrated to reliably match synaptophysin-labelled punctae[51].

For fast imaging of AP-mediated SF-iGluSnFR and Cal-590 fluorescence transients at individual boutons, a spiral-shaped (tornado) scan line was placed over the bouton of interest, as described previously[9] and illustrated in detail in Fig. 6. The spiral tornado mode is a built-in beam scanning option in the Femtonics Femto2D (or Olympus FluoView FV1000) microscope imaging system, allowing direct control over its radius, position, and scanning frequency. The sampling frequency used in the present settings was ~500 Hz, with two-photon excitation at a wavelength $\lambda_x^{2P} = 910$ nm, with the power under objective of 3–5 mW. Through the experimental trials, axonal boutons maintained stable morphological and functional features (nanomolar $Ca^{2+}$ level, evoked $Ca^{2+}$ entry), thus providing direct functional evidence for the experiment-wise absence of photo-toxicity effects.

For simultaneous multi-bouton imaging, the scanning mode was the sequential point-scans over the selected regions of interest (axonal bouton centres), adjusted to a temporal resolution of ~4 ms (~250 Hz rate). The continued cycled series of point-scans over different boutons was subsequently rearranged using a MATLAB algorithm to represent 'pseudo-linescans' associated with individual boutons, as illustrated in Fig. 2b. Following a baseline period, two or four APs, 50 ms apart, were generated using 2 ms positive voltage steps at the cell soma in whole-cell voltage clamp mode ($V_m$ holding at −70 mV).

### FLIM calibration for [$Ca^{2+}$] readout and NTC method.
The Cal-590 [$Ca^{2+}$] calibration protocol was similar to the standard calibration method provided by the Invitrogen $Ca^{2+}$ calibration buffer kit manual. The practical procedure in large part replicated the previously described FLIM calibration protocol for OGB-1[13,14]. In brief, to match the $Ca^{2+}$ buffering dynamics to that of Cal-590 more closely, the standard 10 mM chelating agent EGTA was replaced with 10 mM BAPTA, and the solution constituents were replaced with the experimental intracellular solution (see above). pH was adjusted using KOH, and the KCl concentration was adjusted accordingly, to keep ion constituents in the solution unchanged. The calculated [$Ca^{2+}$] was therefore slightly different from the standard Invitrogen's calibration set and was finely adjusted using Chris Patton's WEBMAXC program (Stanford University, http://www.stanford.edu/~cpatton/webmaxcS.htm). In the control tests, excitation wavelength was varied between 800 and 950 nm and temperature was varied from 20 °C to 37 °C (using a Scientific Systems Design PTC03 in-line heater).

The analysis of fluorescence lifetime data using classical multi-exponential fitting has long been the bottleneck in the experimental throughput. Because the physics that could be inferred from multi-exponential fitting was outside the scope of our studies, we earlier proposed to replace such exponential fitting with a simpler and computationally more economical approach based on a ratiometric method of NTC[13,14]. The fluorescent decay time course was first normalised to its peak value and then integrated (area under curve) over ~3 ns peak-to-peak (Fig. 1a). The resulting value termed NTC was used throughout as an estimator for [$Ca^{2+}$] using an appropriate calibration function: the latter was obtained through direct Cal-590 calibration for intracellular solutions of clamped [$Ca^{2+}$], as explained above, in the microscope imaging system used for experimental measurements (Femto2D, Femtonics, Budapest). The output of Cal-590 lifetime at different clamped [$Ca^{2+}$] values produced almost noiseless decay traces (Fig. 1a) as the photon collection time is not limited by the experimental conditions. In turn, applying the NTC method to these traces and plotting the outcome against the estimated [$Ca^{2+}$] produced a near-perfect fit for logistic function (Fig. 1b).

### Two-photon excitation FLIM of Cal-590 in situ.
With the scanning modes and methods described above, the line-scan data were recorded by both standard analogue integration in Femtonics MES and in TCSPC in Becker and Hickl SPCM using dual HPM-100 hybrid detectors. The two-photon laser source was a Newport-SpectraPhysics Ti:Sapphire MaiTai laser pulsing at 80 Mhz, with a pulse width of ~220 fs and a wavelength of 910 nm optimised for Cal-590 excitation.

FLIM linescan data were collected as previously described[13,14] and stored as 5D-tensors ($t,x,y,z,T$) where $t$ and $T$ stand for the lifetime decay timing and the (global) experiment-wise timing, respectively. The corresponding off-line data sets were analysed using the custom-written analysis software (https://github.com/

zhengkaiyu/FIMAS). For bouton-average Cal-590 FLIM measurements, the $x$, $y$, and $z$ data sets were summed along their respective axes. In contrast, to obtain FLIM readout maps within individual boutons, the recorded photon counts in all relevant co-ordinates (tornado scanning mode) were summed across trials ($T$) to produce count numbers sufficiently high for the reliable NTC measure.

**Quantal analyses of evoked SF-iGluSnFR signals.** Optical glutamate imaging provides a unique opportunity to monitor release activity at individual identified synapses in a non-invasive manner. Similar to the classical electrophysiological studies of synaptic transmission, in this case the quantal size and content of neurotransmitter discharges can be assessed using frequency histograms reflecting trial-to-trial fluctuations of the signal amplitude[21]. In such histograms, amplitude fluctuations around zero (failure) responses reflect the signal noise inherent to the particular recording settings. Establishing the noise fluctuation using a straightforward unsupervised Gaussian fitting (with one free parameter, dispersion $\sigma$; and two fixed parameters, centre $x = 0$ and intersect $y = 0$) will thus constrain (approximately) the key noise parameter $\sigma$, which should apply to all quantal sizes (assuming linear signal scale). In the present study, once $\sigma$ has been constrained, we set up initial parameter values (peak centres $x_i$), their margins (±15% of the mean), $\sigma_i$ margins (±15% of the mean), and employed the unsupervised Levenberg–Marquardt iteration algorithm for multi-Gaussian fitting (GaussAmp function; Origin, OriginLab). The outcome produced individual Gaussians that correspond to the peaks of quantal amplitude and the summated distribution (Figs. 2d and 3g; Supplementary Figure 2f).

**Nanoscopic localisation of glutamate release and $Ca^{2+}$ entry.** In individual axonal boutons, the heat maps of SF-iGluSnFR.A184V signal collected with tornado scans at ~500 Hz were averaged across individual trials along the corresponding time points in the trial. For the period AP-evoked activity (four events 50 ms apart), the maps were plotted as $\Delta F/F$ signal readouts with the basal signal $F$ calculated over the 300-ms prepulse interval. A similar procedure was applied to the Cal-590 FLIM readout signal, with heat maps of the baseline signal reporting resting $[Ca^{2+}]$, and with the heatmaps during the period of AP-evoked activity reporting AP-evoked $[Ca^{2+}]$ increments.

Spatial localisation of the glutamate release site was carried out in the average $\Delta F/F$ SF-iGluSnFR.A184V heat maps (collected during the period between first AP and 15 ms after the fourth AP). The maps represent averaged signals between the time points 2 ms before and 10 ms after the AP-induced fluorescence intensity peaks, thus excluding the values in troughs of the signal waveform. Because the registered glutamate signal hotspot shapes were generally asymmetrical (due to the irregular shape of the boutons and stereological biases), to localise the estimated centres we first used spatial filtering (~100 nm range) and registered the area that has top 5% brightness in the hotspot; in the case of symmetrical hotspots, this procedure generates exactly the same outcome as the standard Gaussian PSF-based localisation.

## Data availability

The data sets generated during and/or analysed during the current study are available from the corresponding author upon reasonable request.

## Code availability

The off-line FLIM data analysis software is available at https://github.com/zhengkaiyu/FIMAS.

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

## Acknowledgements

This work was supported by the Wellcome Trust Principal Fellowship, European Research Council Advanced Grant, Medical Research Council, Biology and Bio-technology Research Council (all UK) and EXTRABRAIN Marie Curie Action (European Commission). The authors thank Mayeul Collot, University of Strasbourg for generous donation of the Calcium Ruby-nano.

## Author contributions

T.P.J. designed and carried out imaging experiments and data analyses; K.Z. designed and implemented time-resolved imaging methods and data analysis algorithms; N.C. carried out slice preparations, transfection, and molecular biology protocols; J.S.M. and L.L.L. developed and provided SF-iGluSnFR glutamate sensor variants; T.P.J., K.Z. and D.A.R. analysed the data; D.A.R. narrated the study, carried out data analyses, and wrote the draft; all authors contributed to manuscript writing.

## Additional information

**Competing interests:** The authors declare no competing interests.

