## [Peer Review File · Nature Communications]

Reviewers' comments:

Reviewer #1 (Remarks to the Author):

General comments on the manuscript:

In this study, Jensen et al describe a new methodological approach to study presynaptic release machinery, combining FLIM and optical quantal analysis with a red-shifted Ca²⁺ indicator, Cal-590, and a green glutamate sensor, SF-iGluSnFR.A184S, in organotypic brain slices. Instead of the single color experiments with OGB-1 and iGluSnFR the authors described in a previous publication (Jensen et al., Cell Calcium 2017), the use of Cal-590 allow them to perform dual color experiments.

While I appreciate the utility of employing intensity-independent and concurrent imaging of presynaptic Ca²⁺ and glutamate release to resolve the structure-function relationship of presynapses, the combination of the two approaches seems to be only an incremental advance, considering previous studies from the same group on FLIM Ca²⁺ imaging, and contributions from other groups employing iGluSnFR in the study of presynaptic release.

Most importantly, the authors provide insufficient experimental evidence to support the biological conclusions presented in the manuscript related to presynaptic release probability, inter-bouton variability, use-dependent plasticity, and the nanoscopic colocalization of Ca²⁺ entry and glutamate release sites, referring too often to the need to perform further separate studies. In order for the study to be relevant, the relationship between calcium transients, glutamate release and synaptic plasticity must be investigated. Additionally, in the described results, it is not clear how many experiments have been made and most figures show repetitive trials on the same axons, which weakens the study. In my opinion, this manuscript presents a nice preliminary work for further study on the presynapse physiology. For instance, the authors end their abstract about the nanoscopic colocalisation of presynaptic calcium entry and glutamate release but this has been hardly shown in the paper on a single bouton and require further investigation.

Specific comments on the manuscript:

1. The statement that this study reveals 'a fundamental unknown in modern neurobiology' is a substantial overstatement considering the weak analysis of biological evidence provided in the manuscript. (p.2, Fig. 3)
2. This is presented as a study using a novel iGluSNFR variant, SF-iGluSnFR.A184S, however this novel variant seems to be used only in Figure 1, whereas the authors use SF-iGluSnFR.A184V for the remaining experiments. The manuscript lacks a clear rationale for the use of A184S and A184V variants, as well as a description of the differential affinity and kinetics of the variants. (p. 4)
3. The manuscript lacks a description of how quantal release of glutamate is measured with iGluSnFR. (p.6)
4. The results presented in the figures seem to reflect multiple trials/sweeps obtained from a single neuron, which, if this is the case, is clearly insufficient to support any of the biological conclusions presented. The results and figure legends are missing a clear description of the number of individual experiments performed.
5. Figure legend 1: the authors state that the traces indicate paired-pulse facilitation, however, from existing literature it seems that wild-type iGluSnFR signals are too slow to accurately track high frequency transmission, resulting in signal summation that cannot be accurately represented without deconvolution. Considering that the authors employ the SF-iGluSnFR.A184S variant with a higher affinity and slower off rate than the wild-type A184V, it seems this construct is not well suited to study use-dependent presynaptic release dynamics. (p. 22)
6. The discussion reads more like an introduction, and much of the information here would be better suited to be presented earlier in the introduction and results sections to better develop and support the logic of inquiry behind this study. (p.9-13)
7. In the discussion section 'Nanoscopic co-localisation of presynaptic glutamate release and Ca²⁺ entry', the authors must expand the presentation of their findings relative to current concepts in the field. In particular, there is an insufficient discussion of highly relevant topics including (1) protein-protein interactions linking presynaptic Ca²⁺ channels and synaptic vesicle proteins that

couple glutamate release to Ca²⁺ influx, (2) the mobility of presynaptic Ca²⁺ channels and the tight vs loose coupling of Ca²⁺ channels and release machinery during plasticity, and (3) the nanostructure of presynaptic active zones and the trans-synaptic alignment of pre- and post-synaptic specializations. (p.12-13).

8. Methods: 37 °C seems a high temperature for the long-term maintenance of organotypic slices. (p. 14)

9. Methods: the duration of the positive voltage step should be specified. (p. 15)

10. Fig. 1c seems to be a composite image – is this a planar projection as in Fig. 2a? This should be specified.

11. In Fig. 2a there seems to be an extra orange arrow near the neuron soma.

12. Fig 2a: The authors show “two main axonal branches” (p. 22), however as one is filled effectively with Cal-590 and the other is not, it seems that the other branch indicated by the white arrows may instead be an axon from another iGluSnFR-transfected neuron.

Reviewer #2 (Remarks to the Author):

Multiplex imaging of quantal glutamate release and presynaptic Ca²⁺ at multiple synapses in situ

This manuscript presents a significant advance in both technology and the application thereof towards understanding a fundamental process in neuroscience: How individual quanta of glutamate release is coupled to presynaptic calcium. By developing the use of CAL-590 as a novel fluorescence lifetime-based calcium imaging probe coupled to the emerging ‘tornado FLIM’ imaging technique, and introducing iGluSnFR as a glutamate probe, the authors present a viable optical method to investigate glutamate/calcium dynamics in vivo, which in theory could pave the way for coupling behavior and/or cognition to neurotransmitter release at individual synapses with (relatively) non-invasive methods. Furthermore, their advanced image acquisition and analysis methodologies apparently facilitate sub-diffraction imaging, allowing them to probe the colocalization of calcium and glutamate at the presynaptic terminal with high resolution. In short, this work is a tour de force that has all the hallmarks of a high impact paper worthy of publication in Nature Communications. I believe this manuscript should be revised before final publication, however, as *some* of the results appear to be either underdeveloped or insufficiently explained/justified. Each novel approach presented here (and there are several!) needs to have strong validation/controls for rigor’s sake. The FLIM data is sufficient, but most of the other methods presented are on more questionable ground, as highlighted in my “running commentary” below.

Figure 1 comments:

The evaluation of CAL-590 FLIM properties appears to be well-thought out and is clearly explained. In contrast, the iGluSnFR results appear thin. For one, I was unable to find anything in the methods or main text detailing the n-values (e.g. number of biological replicates, number of cells, number of synapses) used for these experiments. I am sympathetic to the amount of effort and time it takes to generate these kinds of data, but without at least 3 biological replicates (not even “replicates”, per se – just 3 different examples), one becomes skeptical of the reported results. There is no explanation for how iGluSnFR signal is converted to quantal release probabilities. How were released quanta defined as “detected”? As currently written, these experiments would be difficult for another lab to replicate.

Figure 2 comments:

It is not clear to me that the results in Fig. 2e-f are “beyond the scope of the present...study...”. Yes, it is true that the relationship between neighboring boutons’ calcium dynamics is an entire study in of itself, beyond the scope of this paper. However, this result raises the interesting question as to whether the FLIM readout is affected by the experimental setup. The controls for

calcium buffering are a good start (although they again highlight the lack of biological replicates), but the notable lack of any kind of biological replicates for this type of measurement on neighboring boutons is concerning. Supplementary Fig. 2 should similarly have more replicates.

Figure 3 comments:

This is the weakest, least developed part of the paper. Claims are made without measurements to back them up. The authors state that noise can be reduced using multiple image exposure, but this also increases blur. No quantification of noise vs. blur is performed. The authors claim to use 'stochastic localisation' for nanoscale localization, somewhat analogous to single molecule localization methods. However, the precision of localization is never even estimated, let alone measured. Given the well-established relationship between the number of photons detected vs. the precision of localization, and that the authors are already counting photons, this should be a relatively simple operation. These precision limits will also inform their bouton boundary measurements, which they depend on for their projected image correction. The projected image correction described in this figure and in supplementary figure 3 is begging for a control measurement using an object of known dimensions. Given that one of the main advances claimed by this manuscript is nanoscopic (co)localization, these fundamental measurements simply must be made. The lack of quantification of colocalization is once again a notable omission that highlights the lack of biological replicates or statistics.

Finally, some kind of control with both/either an object of known dimensions, an alternative plasma membrane label, or correlative imaging would go a long way towards establishing the precision of the authors' imaging setup.

Discussion comments:

Throughout the paper, there are several instances where the authors suggest their findings should motivate future studies, which has the unfortunate effect of causing the reader (in this case, reviewer) to wonder whether some of these experiments shouldn't be done for this paper? One such instance is their suggestion of systematically loading axons with different concentrations of Cal-590 to quantitatively assess its effects on synaptic release.

The discussion once again highlights the lack of quantification of actual resolution due to the lack of any measurements of localization precision. This must be fixed. The authors' suggestion that sub-diffraction localization can be achieved using z-stack image acquisition to find maxima in the z-direction appears naïve. While deconvolution could in theory double the z-resolution, further improvements would require an astigmatism or other advanced optics to achieve superresolution in the z-dimension.

Methods comments:

The lack of any details on laser power or the objective used in the methods is a gross oversight. For all the talk about how dyes may affect synaptic activity, the complete disregard for potential phototoxicity is disappointing, although unfortunately not unique in this field. If the authors will not do any controls to account for phototoxicity, they should at least report the doses delivered to the overall cell as well as the power density at each bouton. These details would be...illuminating.

The tornado imaging of action-potential mediated fluorescence transients is not exactly widespread or well-known. The methods should be fully elaborated in the methods section (instead of the current form which refers the reader to "described further in the text").

"the recorded photon counts were summed...for the NTC measure" highlights the lack of an explanation of how this method really works. While it is touched upon and a reference is provided in the main text, I believe the NTC should be fully clarified in the methods section.

Supplementary Figure 3 should be more thoroughly explained in the methods section.

Reviewer #3 (Remarks to the Author):

The manuscript by Jensen et al. describes a method for simultaneous imaging of calcium and glutamate release at presynaptic boutons. They use these tools to attempt to provide a nanoscale description of calcium domains and release sites. As a paper, and specially as a methods paper, this manuscript is incredibly light on detail. In fact, not only are techniques not well described, there are entire experimental techniques that are not mentioned at all. As it stands, the paper is not suitable for publication. Below are some of the many issues with this paper.

1. The methodology of the in vitro characterisation of Cal-590 (Fig.1a-b and supp Fig.1) is not mentioned in the main text or the materials and methods section. The authors cite previous publications for the methods used. This is not enough, especially for a methods paper. Far worse is the fact that the two papers cited are the wrong ones.

2. The authors use tornado scans over the bouton, imaging Ca for both intensity and for FLIM. Again, methods are incredibly sparse. Are tornado scans altered depending on bouton shape/size? In Fig.1d, why does the t-linescan appear as hot-spots (lines) of calcium signal? I would have expected all point along the tornado line scan path to give a signal (perhaps less so for regions of the outer the ring, which may lie outside the bouton). Is this to do with the size of the tornado used? Also, nowhere in the paper does it mention the composition of the extracellular solution.

3. The authors use one of the new iGluSnFR variants A184S to image release at multiple boutons simultaneously in Fig. 1f-h. This variant has a slow off rate enabling the multi boutons sweep. Here, they revert to using line-scans rather than the tornado scan used above. Although the traces look good and by eye it is possible to see failures, no information is given on how glutamate quanta are defined. Once again, this is not covered in the methods section at all. There is no information on image processing or the threshold for signal/noise which is then used to determine successes/failures and therefore Pr. I would also like to know what the spatial resolution of iGluSnFR. Do they see any signals if they image along an axon at increasing distances from a bouton? How much crosstalk/contamination of signal is there between neighbouring boutons?

4. In Fig. 2, the authors describe imaging of glutamate and Ca simultaneously. Here, they switch to the A184V variant which has faster kinetics than A184S, but don't explicitly mention this in the text. All mentions of kinetics in the main text and the discussion are about A184S (including how multiple heterogeneous synapses in identified circuits can be imaged). This is, quite simply, misleading.

5. For the actual multiplex imaging, data from only two boutons is presented in Fig.2. It is not shown where these boutons are along an axon and it is not clear whether it was possible to image these boutons simultaneously as in Fig1f-h. Given the title mentions 'multiple synapses' – how many can be imaged in this way at once? Again, no methods on how glutamate quanta are defined, although lines representing quanta are drawn in fig 2d. The authors state that 'reassuringly the quantal size remained stable throughout trials', shown in Supp Fig 2. However, in the case of bouton 2 particularly one could argue that the variation spans 2 quantal sizes (0.02 and 0.04). There is no information on what level of variation is considered stable.

6. In Fig. 3 the authors present an extension of the method for nanoscopic localisation. Fluorescent signals are averaged over a number of trials and 'stochastic localisation' using the PSF is used to localise the signals. Stochastic localisation is not covered at all in the methods section, although the principles are illustrated in Fig 3c. Here, it appears that the authors have used simulated data, although it does not say this anywhere in the text. More worryingly, in Fig 3a the overlay image suggests some sub-structure already present in the Cal-590 signal – is this the source of some of the hot-spots observed later?

Glutamate release events are corrected for the projection of a sphere onto a plane which would overestimate the edges, explained in Fig 3d,e and Supp Fig 3. For some boutons this will work well but some boutons are more elongated in structure e.g. bouton 2 in Fig 2 - some discussion on disadvantages/advantages of the correction in this case would be good. There is also no discussion of this method in comparison to other methods for nanoscale localisation of release e.g. pHuse. Did the authors try the sphere corrections algorithm in a model (as in Fig.3c, assuming this is a model). Also, why are the images in Fig.3d-g cropped in such odd shapes?

Response to reviewers' comments:

Reviewer #1 (Remarks to the Author):

We are grateful to the reviewer for careful consideration of the manuscript and their valuable comments. We have now carried out multiple additional experiments, in multiple axonal boutons, to directly address important issues raised in the comments.

General

[...] While I appreciate the utility of employing intensity-independent and concurrent imaging of presynaptic Ca²⁺ and glutamate release to resolve the structure-function relationship of presynapses, the combination of the two approaches seems to be only an incremental advance, considering previous studies from the same group on FLIM Ca²⁺ imaging, and contributions from other groups employing iGluSnFR in the study of presynaptic release.

With all due respect, we are slightly bewildered by this comment. Until now, virtually all fundamental discoveries pertinent to the Ca²⁺-dependent machinery of neurotransmitter release have been made in preparations of giant synapses, in which Ca²⁺ imaging could be combined with patch-clamp recordings. The latter has not been feasible for the vast majority of central circuits (which are equipped with small synapses) whereas simultaneous imaging of Ca²⁺ dynamics and glutamate release has long eluded synaptic physiologists. Multiplexed imaging described here enables one, for the first time, to directly examine the relationship between presynaptic calcium dynamics, glutamate release, and synaptic plasticity in the axonal circuit of interest. This line of exploration would not be achievable by simply comparing groups of synapses with separate readouts for Ca²⁺ and glutamate, as we and others reported earlier: release probability and presynaptic Ca²⁺ dynamics could be highly heterogeneous (and use-dependent) within synaptic populations thus making such 'averaged' comparisons spurious. We have expanded the text to clarify this further.

In addition to the scientific concerns that multiplexed imaging resolves, technically it also doubles the experimental throughput, reducing both cost and animal usage for labs wishing to carry out such experiment.

Most importantly, the authors provide insufficient experimental evidence to support the biological conclusions presented in the manuscript related to presynaptic release probability, inter-bouton variability, use-dependent plasticity, and the nanoscopic colocalization of Ca²⁺ entry and glutamate release sites, referring too often to the need to perform further separate studies. In order for the study to be relevant, the relationship between calcium transients, glutamate release and synaptic plasticity must be investigated.

Whilst our study sought to introduce a methodological advance rather than to reveal a biological mechanism, we agree with the Reviewer and have carried out multiple additional experiments addressing the aforementioned tasks. In brief, we have recorded from further ~30 axonal boutons (1-4 boutons per cell), which has enabled us to perform classical quantal analyses and to reveal the relationships between presynaptic Ca²⁺ entry, basal presynaptic [Ca²⁺], glutamate release, and its short-term plasticity, across the sample. Similarly, we have analysed further ~25 boutons for sub-microscopic localisation of glutamate release and Ca²⁺ entry hotspots, assessed their co-localisation, and estimated the diffuse spread of released glutamate. The manuscript has been revamped, expanded and appended with multiple additional figure panels.

Additionally, in the described results, it is not clear how many experiments have been made and most figures show repetitive trials on the same axons, which weakens the study. In my opinion, this manuscript presents a nice preliminary work for further study on the presynapse physiology. For instance, the authors end their abstract about the nanoscopic colocalisation of presynaptic calcium entry and glutamate release but this has been hardly shown in the paper on a single bouton and require further investigation.

The previous version did indeed focus on a small number of synapses, aiming primarily to demonstrate the methodological advance *per se*. With the many-fold increase of the dataset, we have now been able to arrive at robust quantitative conclusions and statistical inference in our data, on several important functional aspects as mentioned above and described further in our specific replies below.

Specific comments on the manuscript:

1. The statement that this study reveals ‘a fundamental unknown in modern neurobiology’ is a substantial overstatement considering the weak analysis of biological evidence provided in the manuscript. (p.2, Fig. 3)

We have rephrased the relevant statements: our intention was to say that the method *should help reveal* fundamental unknowns.

2. This is presented as a study using a novel iGluSnFR variant, SF-iGluSnFR.A184S, however this novel variant seems to be used only in Figure 1, whereas the authors use SF-iGluSnFR.A184V for the remaining experiments. The manuscript lacks a clear rationale for the use of A184S and A184V variants, as well as a description of the differential affinity and kinetics of the variants. (p. 4).

We have clarified and thoroughly revised our statements, also adding quantitative data on the kinetics of the two sniffer variants (main text, Supplementary Fig. 2a). As for the underlying rationale, we simply found that SF-iGluSnFR.A184S and A184V provide optimal features to answer scientific questions which either cannot answer alone. SF-iGluSnFR.A184S enables simultaneous multi-bouton glutamate imaging and therefore study of inter-bouton heterogeneity with a qualitative description of Ca^{2+} dynamics, as FLIM is not possible with short pixel dwell times. SF-iGluSnFR.A184V/ Cal-590 FLIM however enables a more quantitative understanding of Ca^{2+} and glutamate release dynamics at sequentially recorded individual synapses, and without the possibility of indicator saturation at higher frequency spiking patterns. Thus the two sensors represent complimentary properties for studying glutamate release dynamics. The text has been amended to reflect this.

3. The manuscript lacks a description of how quantal release of glutamate is measured with iGluSnFR. (p.6).

As mentioned, we have now added some standard quantal analysis data. These are represented by the signal amplitude histograms, which were best-fitted with multiple Gaussians using a straightforward optimisation procedure partly constrained by the noise distribution parameters, as described in the Methods and Figures (new Figs. 2, 3, Supplementary Fig. 2).

4. The results presented in the figures seem to reflect multiple trials/sweeps obtained from a single neuron, which, if this is the case, is clearly insufficient to support any of the biological conclusions presented. The results and figure legends are missing a clear description of the number of individual experiments performed.

We have now added data of multiple additional experiments, increasing the number of recorded boutons many-fold (up to 35). This has enabled us to arrive at robust quantitative conclusions and statistical inference in our data, as mentioned above and detailed in the text and multiple new figures.

5. Figure legend 1: the authors state that the traces indicate paired-pulse facilitation, however, from existing literature it seems that wild-type iGluSnFR signals are too slow to accurately track high frequency transmission, resulting in signal summation that cannot be accurately represented without deconvolution. Considering that the authors employ the SF-iGluSnFR.A184S variant with a higher affinity and slower off rate than the wild-type A184V, it seems this construct is not well suited to study use-dependent presynaptic release dynamics. (p. 22)

There must have been a misunderstanding. Fluorescent indicators with a fast on-rate and slow off-rate can robustly report relatively high-frequency spike bursts, as long as the indicator is not nearing saturation. In such cases, spikes are reflected by step-wise fluorescence increments: this approach has long been used in Ca^{2+} imaging (e.g., Smetters-Majewska-Yuste 1999 *Methods* 18: 215) while a similar procedure has been routine in analyses of high-frequency trains of overlapping EPSCs. Our optical quantal analyses indicate no saturation of SF-iGluSnFR.A184S in 20 Hz paired-pulse experiments (we did not use it with longer AP trains), with the quantal content scaling almost linearly (new Figs. 2d, 3g, Supplementary Fig. 2f). Similar to the routine of overlapping synaptic currents, the amplitude readout here is obtained by subtracting the pre-increment signal baseline (~8 ms segment, including its time course where required). This procedure does not involve any deconvolution. The text has been expanded accordingly.

6. The discussion reads more like an introduction, and much of the information here would be better suited to be presented earlier in the introduction and results sections to better develop and support the logic of inquiry behind this study. (p.9-13)

We have revised the Discussion accordingly.

7. In the discussion section 'Nanoscopic co-localisation of presynaptic glutamate release and Ca^{2+} entry', the authors must expand the presentation of their findings relative to current concepts in the field. In particular, there is an insufficient discussion of highly relevant topics including (1) protein-protein interactions linking presynaptic Ca^{2+} channels and synaptic vesicle proteins that couple glutamate release to Ca^{2+} influx, (2) the mobility of presynaptic Ca^{2+} channels and the tight vs loose coupling of Ca^{2+} channels and release machinery during plasticity, and (3) the nanostructure of presynaptic active zones and the trans-synaptic alignment of pre-and post-synaptic specializations. (p.12-13).

Intriguingly, our much expanded data set (23 boutons) in these tests generally suggests no prevalent nanoscopic co-localisation between Ca^{2+} entry and glutamate release (new Fig. 6), thus lending support to the loose-coupling hypothesis. Loose coupling is also consistent with our new observations relating trial-by-trial fluctuations in presynaptic basal Ca^{2+} or Ca^{2+} entry to release efficacy (new Fig. 4). We have expanded the discussion as suggested, although we note that the subtleties of the presynaptic molecular machinery require a dedicated study.

8. Methods: 37 °C seems a high temperature for the long-term maintenance of organotypic slices. (p. 14).

We and our departmental colleagues have been using and sharing organotypic cultures for many years. They have been maintained routinely at 36-37C. Maintenance at 37 degrees is also used in labs far longer established using organotypic cultures for synaptic physiology experiments (Debanne et al 2008 *Nat Protocols* 3: 1559–1568).

9. Methods: the duration of the positive voltage step should be specified. (p. 15)

Specified (2 ms).

10. Fig. 1c seems to be a composite image – is this a planar projection as in Fig. 2a? This should be specified.

Because two-photon excitation occurs only within a focal plane, illustrations have to be constructed as image collages consisting of 2D projections of 10-15 μm deep ROI stacks. We have clarified this in the legend.

11. In Fig. 2a there seems to be an extra orange arrow near the neuron soma.

Removed.

12. Fig 2a: The authors show “two main axonal branches” (p. 22), however as one is filled effectively with Cal-590 and the other is not, it seems that the other branch indicated by the white arrows may instead be an axon from another iGluSnFR-transfected neuron.

Here the z-stack of cell morphology was recorded at the laser intensity optimised for the iGluSnFR channel, which avoids image saturation or phototoxicity across the optical sections. These settings were sub-optimal to reveal deeper sections in the Cal-590 channel which was much dimmer. The traced axon, however, goes up and down in the slice where Cal-590 signal might appear and disappear, but it is still reliably traced in the iGluSnFR channel. In the end, axonal boutons are recorded in a single focal plane where both signals are clear. The bouton shown in the old Fig. 2b (new Fig 3b) was traced directly from the soma along the CA3-CA1 axon labelled by white arrows: this bouton cannot belong to a different cell as it responds to the APs evoked in whole-cell at the soma. To clarify the matter further, we have added an example of the full reconstruction of axonal tracing, from the soma to the recorded boutons (new Fig. 4a). The text has been appended accordingly.

Reviewer #2 (Remarks to the Author):

We are grateful to the reviewer for the careful consideration of the manuscript and their insightful and encouraging comments.

This manuscript presents a significant advance in both technology and the application thereof towards understanding a fundamental process in neuroscience: How individual quanta of glutamate release is coupled to presynaptic calcium. By developing the use of CAL-590 as a novel fluorescence lifetime-based calcium imaging probe coupled to the emerging ‘tornado FLIM’ imaging technique, and introducing iGluSnFR as a glutamate probe, the authors present a viable optical method to investigate glutamate/calcium dynamics in vivo, which in theory could pave the way for coupling behavior and/or cognition to neurotransmitter release at individual synapses with (relatively) non-invasive methods. Furthermore, their advanced image acquisition and analysis methodologies apparently facilitate sub-diffraction imaging, allowing them to probe the colocalization of calcium and glutamate at the presynaptic terminal with high resolution. In short, this work is a tour de force that has all the hallmarks of a high impact paper worthy of publication in Nature Communications. I believe this manuscript should be revised before final publication, however, as *some* of the results appear to be either underdeveloped or insufficiently explained/justified. Each novel approach presented here (and there are several!) needs to have strong validation/controls for rigor’s sake. The FLIM data is sufficient, but most of the other methods presented are on more questionable ground, as highlighted in my “running commentary” below.

We understand the concern and have revised our work thoroughly, as detailed below.

Figure 1 comments:

The evaluation of CAL-590 FLIM properties appears to be well-thought out and is clearly explained. In contrast, the iGluSnFR results appear thin. For one, I was unable to find anything in the methods or main text detailing the n-values (e.g. number of biological replicates, number of cells, number of synapses) used for these experiments. I am sympathetic to the amount of effort and time it takes to generate these kinds of data, but without at least 3 biological replicates (not even “replicates”, per se – just 3 different examples), one becomes skeptical of the reported results. There is no explanation for how iGluSnFR signal is converted to quantal release probabilities. How were released quanta defined as “detected”? As currently written, these experiments would be difficult for another lab to replicate.

The previous version did indeed focus on a very small number of synapses, aiming primarily to demonstrate the methodological advance *per se*. We have now carried out multiple additional experiments by recording from further ~30 axonal boutons (1-4 boutons per axon/cell), This has enabled us to perform classical quantal analyses and to reveal the relationships between presynaptic Ca^{2+} entry, basal presynaptic $[\text{Ca}^{2+}]$, glutamate release, and its short-term plasticity, across the sample. Similarly, we have analysed further ~25 boutons for sub-microscopic localisation of glutamate release and Ca^{2+} entry hotspots, assessed their

co-localisation, and estimated the diffuse spread of released glutamate. The manuscript has been revamped, expanded and appended with multiple additional figure panels.

Figure 2 comments:

It is not clear to me that the results in Fig. 2e-f are “beyond the scope of the present...study...”. Yes, it is true that the relationship between neighboring boutons’ calcium dynamics is an entire study in of itself, beyond the scope of this paper. However, this result raises the interesting question as to whether the FLIM readout is affected by the experimental setup.

We showed previously (Zheng et al 2015 *Neuron* 88: 277, 2018 *Nat Prot* 13:581) that the ratiometric FLIM we have implemented could provide clear practical advantages over the classical multi-exponent fitting, but it does require Cal-590 calibration for $[Ca^{2+}]$ in an individual imaging setup (with a particular instrument response). In our experience, once established the calibration remains perfectly stable in a given optical system, across different preparations: it has to be re-calibrated when new optics elements are introduced or when the laser beam parameters change significantly.

The controls for calcium buffering are a good start (although they again highlight the lack of biological replicates), but the notable lack of any kind of biological replicates for this type of measurement on neighboring boutons is concerning.

We have increased our samples many-fold which enabled us to address the issue of internal FLIM-readout controls more robustly. To gauge Cal-590 FLIM-readout sensitivity to dye concentration and Ca^{2+} buffering conditions in axons, we imaged axonal boutons at relatively short times after whole-cell break-in ($n = 7$ cells), when Cal-590 had not yet equilibrated throughout the axon. Thus, during the 20-22 recording trials (1 min apart) the axonal Cal-590 concentration, hence local Ca^{2+} buffering capacity, continued to rise, up to 2-3-fold, as monitored using total photon count of Cal-590 emission (new Supplementary Fig. 4a). Remarkably, we found that this change had no effect on resting $[Ca^{2+}]$ or spike-evoked Ca^{2+} entry measured using Cal-590 FLIM readout (new Supplementary Fig. 4b, c), in full accord with axonal physiology (and high tolerance of steady-state Ca^{2+} homeostasis to local buffering, Zheng et al 2015 *Neuron* 88: 277). Our second test was to compare the dynamic range of Cal-590 lifetime change in calibration conditions *in vitro* (between min and max $[Ca^{2+}]$ load) with that inside an axon (between low resting $[Ca^{2+}]$ and spike-tetanus-saturated Ca^{2+}). We found good compatibility between the two (Fig. 1a and e). Whilst there is no feasible way to completely rule out all possible concomitants affecting indicator characteristics *in situ* (this applies to all known fluorescent dyes), the above tests must provide reasonable assurance to the method robustness.

Supplementary Fig. 2 should similarly have more replicates.

Our approach has been revised. As explained above, new Supplementary Fig. 4 has data from multiple trials in multiple ($n = 7$) cells.

Figure 3 comments:

This is the weakest, least developed part of the paper. Claims are made without measurements to back them up. The authors state that noise can be reduced using multiple image exposure, but this also increases blur. No quantification of noise vs. blur is performed. The authors claim to use ‘stochastic localisation’ for nanoscale localization, somewhat analogous to single molecule localization methods. However, the precision of localization is never even estimated, let alone measured. Given the well-established relationship between the number of photons detected vs. the precision of localization, and that the authors are already counting photons, this should be a relatively simple operation. These precision limits will also inform their bouton boundary measurements, which they depend on for their projected image correction.

We have now carried out these experiments in 23 synapses, which enabled us to modify our objectives and to drastically simplify our measurement approach, also abandoning some irrelevant or unnecessary claims. Straightforward heat maps of iGluSnFR and FLIM Cal-590 signals in individual boutons (averaged for 22-35 trials per synapse) revealed hotspots of activity in at least a sub-group of synapses (9 out of 23), which were much above noise or blur, without any additional filtering or data treatment (new Fig. 6b-c). This amount of data has in turn enabled us to carry out a simple statistical test, by asking whether the expected distance between recorded hotspots of iGlu and Ca^{2+} is compatible with that between points randomly scattered in a similar geometry (new Fig. 6d). The test shows that the former was significantly smaller than the latter, thus proposing that Ca^{2+} entry tends to occur close to glutamate release rather than randomly occurring within the bouton. At the same time, the tests reveal no consistent co-localisation on the $<0.2 \mu\text{m}$ scale (experimental blur should not affect co-localisation inference because iGlu and Cal-590 signals are co-recorded in space and time).

These results lend support to the hypothesis implicating loose coupling between Ca^{2+} entry and release cite (e.g., Vyleta and Jonas 2014 *Science* 343: 665). Loose coupling is also consistent with our new observations relating trial-by-trial fluctuations in presynaptic basal Ca^{2+} or Ca^{2+} entry to release efficacy (new Fig. 4). We have expanded the discussion as suggested. The reviewer will probably agree that these findings pave the way for trying a similar approach in a STED microscope where the nanoscopic relationships could be revealed further. The text has been amended accordingly, with the possible implications discussed.

The projected image correction described in this figure and in supplementary figure 3 is begging for a control measurement using an object of known dimensions. Given that one of the main advances claimed by this manuscript is nanoscopic (co)localization, these fundamental measurements simply must be made. The lack of quantification of colocalization is once again a notable omission that highlights the lack of biological replicates or statistics. Finally, some kind of control with both/either an object of known dimensions, an alternative plasma membrane label, or correlative imaging would go a long way towards establishing the precision of the authors' imaging setup.

We appreciate this comment and have revised our approach in several ways. Firstly, we focused on the relative fluorescence increment ($\Delta F/F$), rather than absolute (F) measurements of the iGlu signal. Because the $\Delta F/F$ measure is independent of the basal intensity, the $\Delta F/F$ heat maps do not need stereological correction for the latter. Again, in this context blur should have little effect on the co-localisation inference as the iGlu and Cal-590 signals are co-recorded in space and time. Nonetheless, because this correction could still be useful for absolute intensity mapping we explained it in the Supplementary Figures and did carry out a control physical measurement, as requested by the reviewer. For that, we used $\sim 1 \mu\text{m}$ empty spherical microcapsules with the fluorescent dye (TRITC/FITC) encapsulated in their thin polymer shell (we used these capsules in our earlier study, Kopach et al 2018 *Drug Delivery* 25:435) and obtained satisfactory representation of surface mapping from planar projection images (new Supplementary Fig. 5g-i).

Secondly, and perhaps more importantly, we have introduced an approximate stereological correction for geodesic (curvilinear surface) distances projected from a spherical or ellipsoidal surface onto a plane (new Fig. 5c-d, new Supplementary Fig. 5d-f). This correction 'stretches' recorded images (new Fig. 5e-g) and is important for assessing the spread of glutamate across the bouton surface (new Fig. 6a).

Discussion comments:

Throughout the paper, there are several instances where the authors suggest their findings should motivate future studies, which has the unfortunate effect of causing the reader (in this case, reviewer) to wonder whether some of these experiments shouldn't be done for this paper? One such instance is their suggestion of systematically loading axons with different concentrations of Cal-590 to quantitatively assess its effects on synaptic release.

As mentioned above, these tests have been carried out. Firstly, we found that a consistent increase in the presynaptic Cal-590 concentration, up to 2-3-fold, had no effect on basal Ca^{2+} or Ca^{2+} entry signal (new Supplementary Fig. 4a-c). Secondly, cell axons with and without Cal-590 (in the working concentration) showed, perhaps surprisingly, indistinguishable average release probability values (new Supplementary Fig. 4d).

The discussion once again highlights the lack of quantification of actual resolution due to the lack of any measurements of localization precision. This must be fixed.

As we have increased our data sets many-fold, we have now been able to arrive at a number of quantitative conclusions on various aspects of the study. These have been appended throughout the manuscript and illustrations.

The authors' suggestion that sub-diffraction localization can be achieved using z-stack image acquisition to find maxima in the z-direction appears naïve. While deconvolution could in theory double the z-resolution, further improvements would require an astigmatism or other advanced optics to achieve superresolution in the z-dimension.

There must have been a misunderstanding. We did not suggest achieving sub-diffraction resolution in the z axis: this does not appear technically feasible without the optics enabling registration of aberration (such as astigmatism) against z coordinate. We suggested that z - stack imaging could help confirm whether the membrane hotspots visible in the focal plane are located on the 'top' or the 'bottom' part of the imaged bouton. We have removed the related claims.

Methods comments:

The lack of any details on laser power or the objective used in the methods is a gross oversight. For all the talk about how dyes may affect synaptic activity, the complete disregard for potential phototoxicity is disappointing, although unfortunately not unique in this field. If the authors will not do any controls to account for phototoxicity, they should at least report the doses delivered to the overall cell as well as the power density at each bouton. These details would be...illuminating.

We apologise for inadvertently omitting this basic detail (we were among the first in Europe to explore real-time physiological effects of 2PE photo-toxicity in neuronal processes). We routinely use 1-8 mW under the objective depending on preparation, experimental design, and scanning mode. It is noteworthy that successful STED imaging of live brain tissue has currently used ~25 mW power of incident laser light (Tonnesen et al 2018 *Cell* 172:1108). In the present work, having the perfectly stable morphological and functional features of axonal boutons (release probability, nanomolar Ca^{2+} level) provides the best possible functional evidence for the experiment-wise absence of photo-toxicity effects. We have appended the Methods accordingly.

The tornado imaging of action-potential mediated fluorescence transients is not exactly widespread or well-known. The methods should be fully elaborated in the methods section (instead of the current form which refers the reader to "described further in the text").

Tornado mode has been a standard 'uncaging' feature of industrial 2PE scopes for some years. We have added further details to the description.

"the recorded photon counts were summed...for the NTC measure" highlights the lack of an explanation of how this method really works. While it is touched upon and a reference is provided in the main text, I believe the NTC should be fully clarified in the methods section.

The key 'ratiometric calculation' detail has been illustrated in new Fig. 1a, and we have added relevant details to the text. A detailed, step-by-step description of the NTC method, with numerous controls, has been published by us recently in considerable detail (Zheng et al 2015 *Neuron* 88: 277, 2018 *Nat Prot* 13:581).

Supplementary Figure 3 should be more thoroughly explained in the methods section.

The figure has been revised, further explanations added.

Reviewer #3 (Remarks to the Author):

The manuscript by Jensen et al. describes a method for simultaneous imaging of calcium and glutamate release at presynaptic boutons. They use these tools to attempt to provide a nanoscale description of calcium domains and release sites. As a paper, and specially as a methods paper, this manuscript is incredibly light on detail. In fact, not only are techniques not well described, there are entire experimental techniques that are not mentioned at all. As it stands, the paper is not suitable for publication. Below are some of the many issues with this paper.s

We are grateful to the Reviewer for their careful consideration of the manuscript and the incisive comments. We do agree that much further detail should be supplied and explained. Importantly, we have carried out multiple additional experiments increasing our sample sizes many-fold (up to 35 individual boutons, 1-4 boutons per axon). This has enabled us to modify our objectives and to drastically simplify some of our measurement approaches, also abandoning some irrelevant or unnecessary claims, as outline below.

1. The methodology of the *in vitro* characterisation of Cal-590 (Fig.1a-b and supp Fig.1) is not mentioned in the main text or the materials and methods section. The authors cite previous publications for the methods used. This is not enough, especially for a methods paper. Far worse is the fact that the two papers cited are the wrong ones.

We appreciate this comment and apologise for what seems to be the last-moment glitch with the citation tool. Indeed, the FLIM method was described in two previous papers in some considerable detail, including step-by-step instructions for the *in vitro* testing and calibration protocols (Zheng et al 2015 *Neuron* 88: 277, 2018 *Nat Prot* 13:581). We have now expanded the Methods to include some key details, as requested.

2. The authors use tornado scans over the bouton, imaging Ca for both intensity and for FLIM. Again, methods are incredibly sparse. Are tornado scans altered depending on bouton shape/size?

The Tornado linescan is circular, with a changeable diameter. We normally place it to be roughly inscribed into the (oval) bouton profile. New illustrations have been added to clarify this.

In Fig.1d, why does the t-linescan appear as hot-spots (lines) of calcium signal? I would have expected all point along the tornado line scan path to give a signal (perhaps less so for regions of the outer the ring, which may lie outside the bouton). Is this to do with the size of the tornado used?

It is indeed likely because part of the spiral went over the bouton edge, but mainly because of the heterogeneous bouton structure: some parts of the bouton contain / project little or no intracellular indicator. For instance, a sphere would have very little projected volume from near its visible edge. Importantly, these Fig. 1d scans represent Cal-590 intensity F whereas the 'ratiometric' $\Delta F/F$ signal is intensity-independent hence much less heterogeneous, revealing true transient signal hotspots, as further explained below.

Also, nowhere in the paper does it mention the composition of the extracellular solution.

Apologies, this has been added to the Methods (a standard solution we use for slice recordings).

3. The authors use one of the new iGluSnFR variants A184S to image release at multiple boutons simultaneously in Fig. 1f-h. This variant has a slow off rate enabling the multi boutons sweep. Here, they revert to using line-scans rather than the tornado scan used above.

We have clarified our statements, also adding quantitative data on the kinetics of the two indicator variants (main text, Supplementary Fig. 2a). The rationale was that SF-iGluSnFR.A184S provided optimal features for multi-bouton glutamate imaging, something that we tried to achieve with other sensors but with less success. In contrast, the A184V variant was faster hence it did not near saturation upon longer bursts of APs. We have expanded the text to further explain the highly complementary use of the two glutamate indicators.

Although the traces look good and by eye it is possible to see failures, no information is given on how glutamate quanta are defined. Once again, this is not covered in the methods section at all. There is no information on image processing or the threshold for signal/noise which is then used to determine successes/failures and therefore Pr.

This is a legitimate request. We have now demonstrated standard quantal analyses in several synaptic examples, showing the signal amplitude histograms best-fitted with multiple Gaussians using a straightforward optimisation procedure that was partly constrained by the noise distribution parameters (Methods; new Figs. 2, 3, Supplementary Fig. 2). This procedure by virtue assesses the noise threshold and thus the likelihood of false-positive signal detection (shown in histograms by yellow shade). We have expanded the text, figure legends, and the Methods accordingly.

I would also like to know what the spatial resolution of iGluSnFR. Do they see any signals if they image along an axon at increasing distances from a bouton? How much crosstalk/contamination of signal is there between neighbouring boutons?

There must have been a minor misunderstanding. The inherent spatial resolution of the iGluSnFR signal is the effective size of iGluSnFR molecules (point sources). This is blurred by the point spread function (PSF) of the optical system, which lowers resolution to 0.2-0.3 μm in the x-y plane. As for the transient iGluSnFR signal spread, it depends entirely on the extent of extrasynaptic glutamate escape via diffusion (plus PSF blur).

New recordings from multiple boutons have enabled us to determine the typical spread of the glutamate signal along the bouton surface, giving the length constant of 0.547 μm (new Fig. 6a). This is fully consistent with the previous estimates of glutamate spillover (Rusakov et al 1999 *TiNS* 22: 382; Diamond 2002 *Nat Neurosci* 5:291), indicating that glutamate cross-talk between boutons of the same axon (several microns apart) is not biophysically plausible. Indeed, an example of simultaneous imaging in two neighbouring boutons shows to detectable cross-talk (new Fig. 2d).

4. In Fig. 2, the authors describe imaging of glutamate and Ca simultaneously. Here, they switch to the A184V variant which has faster kinetics than A184S, but don't explicitly mention this in the text. All mentions of kinetics in the main text and the discussion are about A184S (including how multiple heterogeneous synapses in identified circuits can be imaged). This is, quite simply, misleading.

We have clarified our statements, also adding quantitative data on the kinetics of the two indicator variants (main text, Supplementary Fig. 2a). The original rationale was that SF-iGluSnFR.A184S provided optimal features for simultaneous multi-bouton glutamate imaging, something that we tried to achieve with other sensors but with less success. Our explanations were therefore overly focused on A184S, as opposed to the already established A184V. In contrast, the faster A184V variant was more suitable for longer AP bursts as it remained far from saturation in such conditions. This has now been explained in greater detail in the revision.

5. For the actual multiplex imaging, data from only two boutons is presented in Fig.2. It is not shown where these boutons are along an axon and it is not clear whether it was possible to image these boutons simultaneously as in Fig1f-h.

To address the latter, we have added an example showing simultaneous multiplex imaging of two boutons with SF-iGluSnFR.A184S and Cal-590 (new Supplementary Fig. 3). While this imaging mode was technically feasible, in current settings it remained sub-optimal for Ca^{2+} monitoring as the scanning employed single axonal spots, hence was prone to relatively high noise. Additionally the shorter dwell times required for imaging multiple boutons reduce the photon count considerably making Cal-590 FLIM imaging difficult.

As for the old Fig. 2 data with SF-iGluSnFR.A184V, we have carried out multiple additional experiments increasing the number of boutons recorded in multiplex mode up to 35, with 1-4 boutons per axon imaged sequentially rather than simultaneously (new Fig. 4a). The increased data pool has enabled us to modify our objectives, to streamline our measurements, and to obtain statistical inference for some key relationships between presynaptic Ca^{2+} and glutamate release (new Fig. 4).

Given the title mentions 'multiple synapses' – how many can be imaged in this way at once?

In the first scenario, we could image glutamate release in multiple boutons simultaneously using the SF-iGluSnFR.A184S variant, as shown by examples in new Fig. 2 and in new Supplementary Fig. 2b-f. At the current imaging settings, the number of such boutons will depend on how many can be found within one focal plane, with the theoretical limit of 5-6 (to ensure sufficient beam dwell time per bouton). As mentioned above, we could also image glutamate release and Ca^{2+} kinetics simultaneously in at least two boutons (new Supplementary Fig. 3).

In the second scenario, we used multiplexed (iGlu- Ca^{2+}) high-resolution imaging with Tornado scans and the faster SF-iGluSnFR.A184V variant to image multiple boutons on the same axon sequentially, rather than simultaneously, as illustrated in Fig. 4a. In our hands, the upper limit for such boutons is 5-6 depending on cell health. We have clarified these aspects in the text.

Again, no methods on how glutamate quanta are defined, although lines representing quanta are drawn in fig 2d. The authors state that 'reassuringly the quantal size remained stable throughout trials', shown in Supp Fig 2. However, in the case of bouton 2 particularly one could argue that the variation spans 2 quantal sizes (0.02 and 0.04). There is no information on what level of variation is considered stable.

As mentioned above, we have now applied our classical quantal analysis routines using the $\Delta F/F$ signal amplitude histograms fitted with multiple Gaussians: this unsupervised procedure was partly constrained by the noise (failure response) parameters (new Figs. 2d, 3g, Supplementary Fig. 2f). Again, this approach by virtue assesses the noise threshold including the likelihood of false-positive signal detection. We have expanded the text, figure legends, and the Methods accordingly.

6. In Fig. 3 the authors present an extension of the method for nanoscopic localisation. Fluorescent signals are averaged over a number of trials and 'stochastic localisation' using the PSF is used to localise the signals. Stochastic localisation is not covered at all in the methods section, although the principles are illustrated in Fig 3c. Here, it appears that the authors have used simulated data, although it does not say this anywhere in the text.

We have now carried out these experiments in 23 synapses, which enabled us to modify our objectives and to drastically simplify our measurement approach, also abandoning some irrelevant or unnecessary claims. Straightforward heat maps of iGluSnFR and Cal-590 FLIM signals in individual boutons (averaged for 22-35 trials per synapse) revealed clear hotspots of activity in at least a sub-group of synapses (9 out of 23), which were much above the noise or blur, without any additional filtering or data treatment (new Fig. 6b-c). This amount of data has enabled us to carry out a simple statistical test, by asking whether the expected distance between recorded hotspots of iGluSnFR and Ca^{2+} is compatible with that between points randomly scattered in a similar geometry (new Fig. 6d). The test shows that former was significantly smaller than the latter, thus proposing that Ca^{2+} entry tends to occur close to glutamate release rather than randomly occurring within the bouton. At the same time, the

tests reveal no general co-localisation on the $<0.2 \mu\text{m}$ scale, which is somewhat unexpected: blur should not affect co-localisation inference as iGlu and Cal-590 signals are co-recorded in space and time.

Clearly, this spatial relationship is important, it raises questions about how the intra-terminal Ca^{2+} spread controls release, about the molecular machinery involved, and how it changes during plasticity events. The text and illustrations have been revised accordingly, with the possible implications discussed.

More worryingly, in Fig 3a the overlay image suggests some sub-structure already present in the Cal-590 signal – is this the source of some of the hot-spots observed later?

The illustration is simply an intensity image with a significant amount of noise. In such images, Cal-590 fluorescence signal will be proportional to the cytoplasm volume occupied by Cal-590. This volume heterogeneity leads to apparent 'hotspots' readily seen in intensity scans, as mentioned above (Fig. 1d, 3b). The relative measure $\Delta F/F$, however, nullifies such heterogeneity whereas FLIM readout (time domain measure) is by definition insensitive to the dye concentration or volume. The hotspots seen in multiplexed imaged (Fig. 5f-g, 6b-c) are therefore not biased by fluctuations in dye concentration or volume.

Glutamate release events are corrected for the projection of a sphere onto a plane which would overestimate the edges, explained in Fig 3d,e and Supp Fig 3. For some boutons this will work well but some boutons are more elongated in structure e.g. bouton 2 in Fig 2 - some discussion on disadvantages/advantages of the correction in this case would be good.

With much more data at hand, this part has been thoroughly revised. Firstly, we focused on the relative fluorescence increment ($\Delta F/F$), rather than absolute (F) measurements of the iGlu signal. Because the $\Delta F/F$ measure is independent of the basal intensity, the $\Delta F/F$ heat maps do not need stereological correction for the latter. More to the point, we have introduced an approximate stereological correction for geodesic (curvilinear surface) distances projected from a spherical or ellipsoidal surface onto a plane (new Fig. 5c-d, new Supplementary Fig. 5d-f). This correction 'stretches' recorded images (new Fig. 5e-g) and is important for assessing the spread of glutamate across bouton surface (new Fig. 6a).

The average heat maps of iGlu and FLIM Cal-590 signals in individual boutons (averaged for 22-35 trials per synapse) revealed clear hotspots of activity in at least a sub-group of synapses (9 out of 23), which were much above the noise or blur, without any additional filtering or data treatment (new Fig. 6b-c). This amount of data has in turn enabled us to carry out a simple statistical test, by asking whether the expected distance between recorded hotspots of iGluSnFR and Ca^{2+} is compatible with that between points arbitrarily scattered in a similar geometry (new Fig. 6d). The test shows that the former was significantly smaller than the latter, thus proposing that Ca^{2+} entry does tend to occur close to glutamate release rather than randomly occurring within the bouton. At the same time, the tests reveal no co-localisation on the $<0.2 \mu\text{m}$ scale.

This result lends support to the hypothesis implicating loose coupling between Ca^{2+} entry and release cite (e.g., Vyleta and Jonas 2014 *Science* 343: 665). Loose coupling is also consistent with our new observations relating trial-by-trial fluctuations in presynaptic basal Ca^{2+} or Ca^{2+} entry to release efficacy (new Fig. 4). We have expanded the discussion as suggested. The text has been amended accordingly, with the possible implications discussed.

There is also no discussion of this method in comparison to other methods for nanoscale localisation of release e.g. pHuse.

We have added a brief discussion pertinent to the pHuse localisation method employing Phluorins in cultured cells.

Did the authors try the sphere corrections algorithm in a model (as in Fig.3c, assuming this is a model).

Yes, although we stopped using this correction for mapping glutamate and Ca^{2+} transients (we focused instead on intensity-independent ratiometric readouts, $\Delta F/F$ or FLIM), it could still be useful for intensity mapping. Therefore, we did carry out a control measurement with a real physical model. For that, we used $\sim 1 \mu\text{m}$ empty spherical microcapsules with the fluorescent dye (TRITC/FITC) encapsulated in their thin polymer shell (we used these capsules in our earlier study, Kopach et al 2018 *Drug Delivery* 25:435) and obtained satisfactory representation of surface mapping from planar projection images: this task has been moved to the Supplementary material (new Supplementary Fig. 5g-i).

Also, why are the images in Fig.3d-g cropped in such odd shapes?

The old image was cropped to roughly follow the bouton shape but we have now thoroughly revised that part, adding new multiple boutons data: here, the heat maps consistently follow the original circular Tornado scans (Fig. 6c).

Reviewers' comments:

Reviewer #1 (Remarks to the Author):

The authors have achieved a remarkable job at revising their MS, providing ample new data and analysis, pushing this new version to high standard. The obtained results are very convincing and exciting and should be of interest to a broad Community.

A minor comment, Fig 4g, why isn't regression shown to show independence ?

Reviewer #2 (Remarks to the Author):

I apologize for the delay in reviewing the paper, as the revision was sent to me shortly before Thanksgiving break and I was traveling.

The authors have properly addressed all of my major concerns quite nicely, and the manuscript is now suitable for publication, in my opinion.

Reviewer #3 (Remarks to the Author):

The resubmitted manuscript has now significantly improved, including more details on the techniques used and on data analysis. However, although these tools described here are certainly useful to assess synaptic release properties, the biological questions that are tackled in this paper remain less clear. I can see the potential for the tools, but the authors have not fully exploited them to answer a biological question.

The authors devote considerable effort to justifying the use of iGluSnFR for measurements of quantal events. The histograms shown do in Figures 2 and 3 suggest that the authors may be able to detect quantal events. As a result, they also calculate release probability (Pr) for each bouton, which is an important measure of bouton function. Surprisingly, they then ignore this measure for the rest of the paper and focus exclusively on the DF/F measures of iGluSnFR, rather than Pr. There may be clear reasons why the authors decided to focus DF/F measures, but they are not explained in the text and, quite frankly, the absence of an explanation raises concerns. Is there a relationship between DF/F amplitude and Pr? One would expect this to be the case, although that is certainly not obvious from the examples given in Fig.2. In relation to this, how is Pr calculated when synapses undergo multi-vesicular release, as shown in Figs.2 and 3?

In my view, the most puzzling measure used in this paper is shown in Fig. 4b and Fig. 3h. If I understood correctly, each point in the graphs on Fig 4b correspond to single iGluSnFR DF/F responses plotted against resting calcium levels. The same type of plot is shown in Fig. 3h, although in this case it is plotted against the Cal-590 DF/F amplitude. Since the authors claim to measure quantal responses with iGluSnFR, the amplitude of the DF/F signal will simply measure the amount of glutamate released by a vesicle. It is not a measure of Pr. It is not clear to me why the authors conclude from this data that neurotransmitter release (and specifically Pr) is dependent on resting calcium. Nor does it make sense to state in Fig 3h 'that release probability at this synapse depends on the trial-by-trial fluctuations of AP-evoked Ca²⁺ entry'. The sum of responses in Fig. 4c and d are also strange ways of measuring release and Pr from boutons. It's unclear to me what all these correlations mean. Are the authors suggesting that quantal content depends on resting calcium levels?

A more meaningful measure is to take either the average DF/F response for each bouton (which I assume is correlated to Pr, although this would have to be shown) or, even better, Pr itself (which

the authors have already shown they can measure). These are the measures of neurotransmitter release that, in my opinion, are informative. In fact, this kind of analysis was carried out for short-term plasticity in Fig. 4f and g, where the average properties of the synapses over multiple trials are related to calcium. Unfortunately, there were very few boutons in this analysis, which makes interpretation of these results troubling. Indeed, this may be the problem with the analysis I propose above and may be the biggest drawback of this paper. Drawing conclusions of synaptic properties from a limited number of boutons that are functionally heterogeneous is an important issue.

As an aside, why do the number of boutons used in each graph in Figure 4 vary? Did the authors not use the same data sets?

Finally, I am unsure what to make of the mapping of submicron hotspots. I like the idea, but I am unsure about the significance of their findings. Again, only 9 boutons could be used and the biological conclusions are not obvious. How do the authors know that these hotspots are real? Could they stain for active zone proteins or calcium channels and see if they co-localise with release or calcium hotspots?

Overall, this paper shows a promising technique that needs to be implemented further to answer a biological question.

Point-to-point reply to Reviewers comments

Reviewer #1 (Remarks to the Author):

The authors have achieved a remarkable job at revising their MS, providing ample new data and analysis, pushing this new version to high standard. The obtained results are very convincing and exciting and should be of interest to a broad Community.

We appreciate careful consideration and positive feedback from the reviewer.

A minor comment, Fig 4g, why isn't regression shown to show independence ?

These data show a large variation of plasticity parameters among tested synapses, which is likely to depend on multiple factors (synaptic structure, channel organisation, functional history, etc.) that would dwarf any Ca^{2+} dependencies. Thus, it would not appear reliable to draw any conclusions from such data regarding the plasticity-vs- Ca^{2+} relationship. Instead, we have added a more relevant data set showing no Ca^{2+} dependence of short-term plasticity in trial-to-trial comparisons within individual synapses (Supplementary Fig. 5). The text has been appended accordingly.

Reviewer #2 (Remarks to the Author):

I apologize for the delay in reviewing the paper, as the revision was sent to me shortly before Thanksgiving break and I was traveling. The authors have properly addressed all of my major concerns quite nicely, and the manuscript is now suitable for publication, in my opinion.

We appreciate careful consideration and positive feedback from the reviewer.

Reviewer #3 (Remarks to the Author):

The resubmitted manuscript has now significantly improved, including more details on the techniques used and on data analysis. However, although these tools described here are certainly useful to assess synaptic release properties, the biological questions that are tackled in this paper remain less clear. I can see the potential for the tools, but the authors have not fully exploited them to answer a biological question.

We appreciate careful consideration of our manuscript and valuable comments. We are however not entirely certain how to interpret the 'biological question' comment. Indeed, the present study was primarily designed to demonstrate a new method and its gnostic potential. However, our observations do shed light on some important biological questions that have long eluded synaptic physiologists. We show that: (a) glutamate release at small central synapses fluctuates over time, trial-to-trial, reflecting variations in evoked Ca^{2+} entry and the resting presynaptic $[\text{Ca}^{2+}]$, (b) short-term plasticity does not depend on Ca^{2+} entry or resting presynaptic $[\text{Ca}^{2+}]$, (c) in individual axonal boutons, glutamate release tends to concentrate at one hotspot whereas Ca^{2+} entry does not, pointing to loose coupling between the two, and (d) glutamate escapes from its release site (before being taken up) with a length constant of $\sim 0.55 \mu\text{m}$. Most of these issues have long been points of contention, which we hope our data can help to resolve.

The authors devote considerable effort to justifying the use of iGluSnFR for measurements of quantal events. The histograms shown do in Figures 2 and 3 suggest that the authors may be able to detect quantal events. As a result, they also calculate

release probability (P_r) for each bouton, which is an important measure of bouton function. Surprisingly, they then ignore this measure for the rest of the paper and focus exclusively on the DF/F measures of iGluSnFR, rather than P_r . There may be clear reasons why the authors decided to focus DF/F measures, but they are not explained in the text and, quite frankly, the absence of an explanation raises concerns.

Is there a relationship between DF/F amplitude and P_r ? One would expect this to be the case, although that is certainly not obvious from the examples given in Fig.2. In relation to this, how is P_r calculated when synapses undergo multi-vesicular release, as shown in Figs.2 and 3?

There must have been some misunderstanding pertaining to the relationship between average release probability P_r , 'instantaneous' vesicle release probability $P_{r/v}$, and presynaptic Ca^{2+} , probably due to the insufficient explanations in the text. The Reviewer is of course correct that P_r is a basic, well-established indicator of synaptic efficacy. P_r is an essential measure in monitoring average synaptic strength and its evolution during physiological changes, or in comparing identifiable synaptic populations. However, P_r values are much less usable when attempting to understand how presynaptic Ca^{2+} controls neurotransmitter release. This is for several important reasons, as explained below.

Firstly, P_r varies among synapses enormously: in our case, between 0.05-1, or ~20-fold (Fig. 5d). This variability depends heavily on the multiple factors shaping synaptic identity, such as the active zone size and composition, size and position of vesicle pool(s), activity history, functional state, Ca^{2+} channel composition and distribution, etc. Whilst presynaptic Ca^{2+} homeostasis could also contribute to the inter-synaptic variability of P_r , the above (poorly controlled) factors usually dwarf its influence. In other words, attempts to draw conclusions about the relationship between presynaptic Ca^{2+} and P_r by comparing them *among* synapses are likely to produce spurious results. That was the reason why the classical studies in giant synapses (accessible to patch-clamp) focused on manipulating Ca^{2+} homeostasis in *individual synapses*: to keep constant all other synapse-specific concomitants of release efficacy. Our multiplexed imaging technique enables this type of probing in small central synapses.

Secondly, classical P_r is only a basic estimator of synaptic efficacy, calculated as the average release success rate over a number of trials. Assessing P_r does not require quantal analyses, just a reliable distinction between release successes (mono- or multi-vesicular) and failures. Critically, P_r values provide us with little knowledge of what is the actual probability of release at any given trial, or the 'instantaneous' vesicle release probability $P_{r/v}$, and its temporal evolution (e.g., Park et al, 2012, Science 335: 1362).

Finally, as correctly noted by the Reviewer, P_r values do not reflect the fact that multi-vesicular release corresponds to stronger synaptic input if compared with one-vesicle release.

Therefore, to gauge synaptic efficacy more directly we have focused on the amount of released glutamate reported trial-to-trial by either the DF/F iGluSnFR signal or the quantal content readout, which represent a statistical readout of $P_{r/v}$, as further explained below.

In my view, the most puzzling measure used in this paper is shown in Fig. 4b and Fig. 3h. If I understood correctly, each point in the graphs on Fig 4b correspond to single iGluSnFR DF/F responses plotted against resting calcium levels. The same type of plot is shown in Fig. 3h, although in this case it is plotted against the Cal-590 DF/F amplitude.

Since the authors claim to measure quantal responses with iGluSnFR, the amplitude of the DF/F signal will simply measure the amount of glutamate released by a vesicle. It is not a measure of P_r .

There must have been a misunderstanding. The DF/F iGluSnFR signal scales with the amount of released glutamate, which fluctuates stochastically trial-to-trial, depending on the number of released vesicles (i.e., quantal content), just as the EPSC amplitudes fluctuate in a well-clamped postsynaptic cell. Because at each given trial, glutamate release signal reflects statistically the instantaneous vesicle release probability $P_{r/V}$ {Park, 2012 #9760; Volynski, 2006 #3605; McGuinness, 2010 #9778), the observed correlations convey the strong dependence of $P_{r/V}$ on the trial-to-trial fluctuations in AP-evoked Ca^{2+} entry. This dependence has not been shown previously.

Considering terminology, we agree with the reviewer that it might be confusing to directly refer to the DF/F data as quantal responses. In addition, the DF/F iGluSnFR signal per released vesicle could vary among synapses due to varying experimental settings. Although this variation should not bias data regression (correlations) in a combined sample, it blurs the notion of the uniform quantal amplitude. To avoid such confusion, we have therefore systematically implemented quantal analyses on all recorded synapses replacing raw DF/F values with quantal content values, thus 'normalising' release data across synapses (Fig. 3h, Fig. 4b-d). As expected, correlation estimates in such data were similar to those in the original raw DF/F data (which have now been moved to the new Supplementary Fig. 4).

It is not clear to me why the authors conclude from this data that neurotransmitter release (and specifically P_r) is dependent on resting calcium. Nor does it make sense to state in Fig 3h 'that release probability at this synapse depends on the trial-by-trial fluctuations of AP-evoked Ca^{2+} entry'.

We have now made clear, throughout the text, the distinction between average release probability P_r and instantaneous (vesicular) release probability $P_{r/V}$, which is what the Reviewer refers to.

The sum of responses in Fig. 4c and d are also strange ways of measuring release and P_r from boutons. It's unclear to me what all these correlations mean. Are the authors suggesting that quantal content depends on resting calcium levels?

Indeed, these data explicitly suggest that at each trial the quantal content of glutamate release (which depends on $P_{r/V}$), be it a single or short-train response depends on the concurrent values of resting $[Ca^{2+}]$ and Ca^{2+} entry, trial-to-trial, in individual synapses.

Again, the *average* quantal content (or average P_r) also varies *among* synapses but such variability depends on multiple, poorly controlled factors, as explained above. Assessing such factors is beyond the scope of the present study.

A more meaningful measure is to take either the average DF/F response for each bouton (which I assume is correlated to P_r , although this would have to be shown) or, even better, P_r itself (which the authors have already shown they can measure). These are the measures of neurotransmitter release that, in my opinion, are informative.

As explained above, a key subject of our study was instantaneous vesicular release probability $P_{r/V}$ rather than average P_r per synapse.

In fact, this kind of analysis was carried out for short-term plasticity in Fig. 4f and g, where the average properties of the synapses over multiple trials are related to calcium. Unfortunately, there were very few boutons in this analysis, which makes interpretation of

these results troubling. Indeed, this may be the problem with the analysis I propose may be the biggest drawback of this paper. Drawing conclusions of synaptic properties from a limited number of boutons that are functionally heterogeneous is an important issue.

We hope our explanations above, and the corresponding revision in the manuscript, help to clarify the notion and the specific relevance of P_r and $P_{r/V}$ values. As for the short-term plasticity, again, Fig. 4g only illustrates average comparisons data among synapses. The more relevant trial-to-trial plasticity data (Fig. 4e, new Supplementary Fig. 5) suggest less spuriously that such plasticity independent of presynaptic Ca^{2+} homeostasis. The text has been expanded accordingly.

As an aside, why do the number of boutons used in each graph in Figure 4 vary? Did the authors not use the same data sets?

In original Fig. 4d and 4g (new Figs. 5d, e), the number of tested boutons was smaller because, historically, only two (rather than four) APs were applied in some tests: the corresponding boutons were consequently excluded from the 'four-AP' data. The text has been appended accordingly. In Fig. 4e, number 427 was mistyped instead of 426.

Finally, I am unsure what to make of the mapping of submicron hotspots. I like the idea, but I am unsure about the significance of their findings. Again, only 9 boutons could be used and the biological conclusions are not obvious. How do the authors know that these hotspots are real?

There must have been a misunderstanding or perhaps an oversight. In all 23 boutons, heat maps of iGluSnFR show a clear hotspot peak (single in 20 boutons and double in 3 boutons, as illustrated in Fig. 7a). This is quantitatively shown by a >10-fold drop in the iGluSnFR signal brightness profile, for all 23 boutons, within 0.5-1 μm from the peak (Fig. 7b). These data also indicate that the imaging system and the protocol are fully capable to detect such hotspots. At the same time, Ca^{2+} signal collected simultaneously in similar conditions appears disperse: in nine synapses where individual Ca^{2+} entry hotspots were discernible (>2SD above the noise, Fig. 7c), they did not coincide with glutamate peaks. Thus, multiplexed recordings of glutamate release and Ca^{2+} entry obtained in 23 synapses lend support to the loose-coupling hypothesis, as detailed in the text.

Could they stain for active zone proteins or calcium channels and see if they co-localise with release or calcium hotspots?

We appreciate this suggestion, which is not, unfortunately, technically feasible. Firstly, active zone proteins do not have to necessarily coincide with one particularly active release site. Secondly, live Ca^{2+} channel label preserving channel function have not been available, to our knowledge.

Overall, this paper shows a promising technique that needs to be implemented further to answer a biological question.

We are grateful to the reviewer for this encouraging comment.

REVIEWERS' COMMENTS:

Reviewer #3 (Remarks to the Author):

The authors have dealt with my concerns and provided a detailed logic of their findings. Although some aspects of their findings remain unclear to me, the tools described here will certainly be of use to the community. I support its publication.